# Image Clustering via the Principle of Rate Reduction in the Age of Pre-trained Models

**Tianzhe Chu**[1,2*]    **Shengbang Tong**[1*]    **Tianjiao Ding**[3*]    **Xili Dai**[4]

**Benjamin D. Haeffele**[5]    **René Vidal**[3]    **Yi Ma**[1,6]

## Abstract

The advent of large pre-trained models has brought about a paradigm shift in both visual representation learning and natural language processing. However, clustering unlabeled images, as a fundamental and classic machine learning problem, still lacks an effective solution, particularly for large-scale datasets. In this paper, we propose a novel image clustering pipeline that leverages the powerful feature representation of large pre-trained models such as CLIP and cluster images effectively and efficiently at scale. We first developed a novel algorithm to estimate the number of clusters in a given dataset. We then show that the pre-trained features are significantly more structured by further optimizing the rate reduction objective. The resulting features may significantly improve the clustering accuracy, e.g., from 57% to 66% on ImageNet-1k. Furthermore, by leveraging CLIP's multimodality bridge between image and text, we develop a simple yet effective self-labeling algorithm that produces meaningful captions for the clusters. Through extensive experiments, we show that our pipeline works well on standard datasets such as CIFAR-10, CIFAR-100, and ImageNet-1k. It also extends to datasets that are not curated for clustering, such as LAION-Aesthetics and WikiArts. We released the code in `https://github.com/LeslieTrue/CPP`.

## 1 Motivation

Clustering is a fundamental problem in machine learning, with many common methods emerging as early as the 1950s (Lloyd, 1957; Forgey, 1965; Jancey, 1966; McQueen, 1967) along with numerous modern developments. Nevertheless, there is a significant discrepancy separating the recent advance of *large-scale methods* from that of *clustering performance*. Namely, there are datasets with millions (Russakovsky et al., 2015) or even billions (Schuhmann et al., 2022) of images and thousands of classes and classification methods with close to 100% accuracy, yet existing clustering approaches typically either fail on natural images (Bradley et al., 1996; Bahmani et al., 2012; Souvenir & Pless, 2005; Elhamifar & Vidal, 2011; 2013; Patel & Vidal, 2014; Lu et al., 2012; Liu et al., 2013; Heckel & Bölcskei, 2015; You et al., 2016a; Tsakiris & Vidal, 2017), or have been tested only with datasets of a small number of clusters ($\sim 10^2$) and images ($\sim 10^5$) (Park et al., 2021; Li et al., 2021; Yaling Tao, 2021; Deshmukh et al., 2021; Niu & Wang, 2021; Li et al., 2022; Sadeghi et al., 2022) with some exceptions (Van Gansbeke et al., 2020; Adaloglou et al., 2023). So far, few methods have achieved above 50% clustering accuracy on ImageNet-1k or TinyImageNet-200: e.g., Van Gansbeke et al. (2020); Li et al. (2021); Niu & Wang (2021); Sadeghi et al. (2022); Ding et al. (2023) all achieve accuracy smaller than 50%.

Classic clustering methods often build on assumptions about the geometry of data from each cluster, such as modeling each cluster as a centroid (Lloyd, 1957; Forgey, 1965; Bradley et al., 1996; Arthur & Vassilvitskii, 2006), a linear or affine subspace of low (Elhamifar & Vidal, 2013; Heckel & Bölcskei, 2015; You et al., 2016a) or high dimension (Tsakiris & Vidal, 2017; Ding et al., 2021; 2024), a manifold from known families (Patel & Vidal, 2014), sampled densely (Souvenir & Pless, 2005), or locally approximated by affine subspaces (Elhamifar & Vidal, 2011). Although effective on relatively simple datasets such as COIL (Nene et al., 1996) or MNIST (LeCun, 1998), these methods are either *not accurate* (the geometric assumptions are drastically violated) or *not scalable* (computing a neighborhood graph is expensive) when confronted with more complicated or large-scale datasets such as CIFAR (Krizhevsky et al., 2009) and ImageNet (Russakovsky et al., 2015).

[*]Equal contribution, [1]University of California, Berkeley, [2]ShanghaiTech University, [3]University of Pennsylvania, [4]Hong Kong University of Science and Technology (Guangzhou), [5]Johns Hopkins University, [6]Hong Kong University

Key to the recent advance in clustering is *teaching an old dog new tricks*, i.e., using deep networks to learn features with desirable geometric properties that can be used for clustering (and other downstream tasks). Recent clustering pipelines (Van Gansbeke et al., 2020; Ding et al., 2023; Niu et al., 2022) proceed by 2 steps: i) learning an initial representation by self-supervised learning, e.g., the joint-embedding approaches (Chen et al., 2020; He et al., 2020; Tong et al., 2023), and ii) gradually refining the representation and clustering membership after the initialization. One family of methods (Li et al., 2022; Ding et al., 2023) base their design on the principle of Maximal Coding Rate Reduction (MCR$^2$, Yu et al. (2020)). In short, these methods aim to learn a representation such that features within the same cluster tend to span a low-dimensional subspace (i.e., within-cluster diverse), and subspaces from different clusters tend to be orthogonal (between-cluster discriminative). Promising as it may seem, clustering methods initialized by self-supervised learning highly depends on the initial representation, and thus clustering performance is far from the supervised classification baselines on CIFAR-100 or ImageNet-1k.

In parallel with these developments, the advent of large-scale pre-trained models such as Contrastive Language-Image Pre-training (CLIP, Radford et al. (2021)) and Self-Distillation with No Labels (DINO, Caron et al. (2021); Oquab et al. (2023)) have showcased an impressive capacity to learn rich representations from a wide variety of datasets. In particular, CLIP is trained by pairing a diverse set of images with natural language descriptions. It has been shown to serve as a foundation model that scales up to training data, making it highly suitable for tasks that require a nuanced understanding of visual information.

**Contributions.** To address the challenges inherent in clustering large-scale and uncurated data and really push the limit of clustering, we leverage the advance in both pre-trained models and principled clustering approaches to develop a novel pipeline, named CPP (**C**lustering via the **P**rinciple of rate reduction and **P**retrained models). In particular, this paper makes the following contributions:

1. We propose to integrate the powerful image encoder from CLIP into the clustering framework MLC, and demonstrate that such a combination leads to state-of-the-art clustering performance on standard datasets CIFAR-10 and -20 (Krizhevsky et al., 2009), and further on large-scale datasets CIFAR-100 and ImageNet-1k (Russakovsky et al., 2015); the latter two are often ignored by prior works.

2. While prior clustering methods typically assume the number of clusters is given, it is often unknown for large and uncurated datasets. Therefore, we provide a model selection mechanism suitable for MLC that estimates the optimal number of clusters *without any costly retraining*. We validate this mechanism on CIFAR-10, -100, and further apply it to MS-COCO (Lin et al., 2014), LAION-Aesthetic (Schuhmann et al., 2022), and WikiArt (Saleh & Elgammal, 2015).

3. To further label the obtained clusters with semantic descriptions that can be comprehended by a human, we propose a simple yet effective self-labeling algorithm utilizing the vision-text binding provided by CLIP. We show that the entire pipeline yields semantically meaningful clusters on MS-COCO, LAION-Aesthetic, and WikiArt.

## 2 RELATED WORK

In this section, we broadly review some of the work that has inspired our research. We begin with the recent progress in pre-trained models, and then discuss the recent trend of image clustering via pre-trained models.

**Pre-trained Models in Vision.** In recent years, we have witnessed the rapid development of vision pre-trained models in the field. Pure vision models have benefited from advances in joint-embedding self-supervised learning (Chen et al., 2020; He et al., 2020; Bardes et al., 2021; Caron et al., 2021; Grill et al., 2020) and masked self-supervised learning (He et al., 2022; Zhou et al., 2021; Bao et al., 2021). These models have shown great promise in learning semantically meaningful features on large scale of datasets.

Another line of work focuses on learning meaningful representation from images with the guidance of text information. Inspired by the progress in contrastive learning (Chen et al., 2020), CLIP proposes contrastive language-image pre-training. The work demonstrates incredible scalability and very strong performance. Follow-up reproduction openCLIP (Ilharco et al., 2021) has verified that the method's scalability with the largest vision transformer model (Dosovitskiy et al., 2020) and the

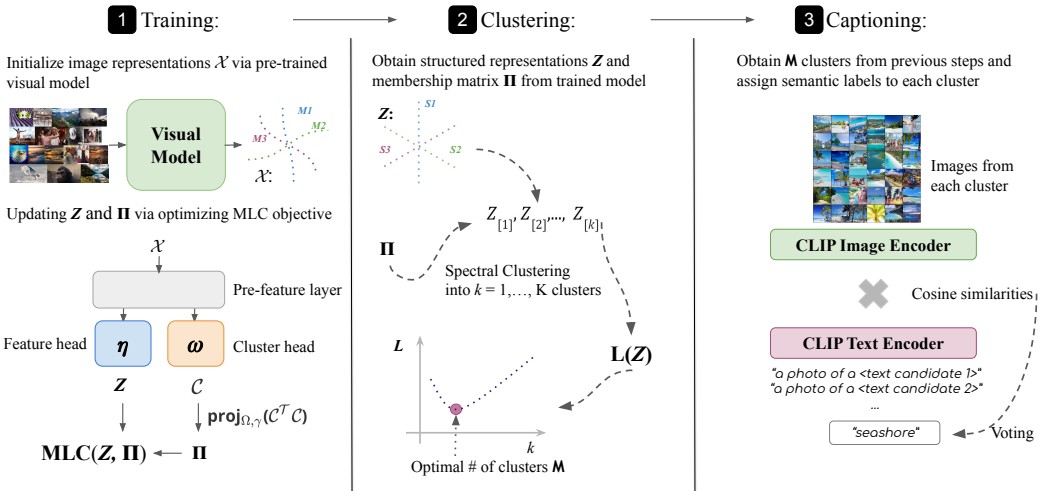

Figure 1: **Overall pipeline of CPP**. *Left*: In the training stage, CPP initializes the features $\mathcal{Z}$ and cluster membership $\Pi$ from a large pre-trained model, and updates $\mathcal{Z}$ and $\Pi$ by optimizing the (MLC) objective. *Middle*: Once training is done, CPP selects the optimal number of clusters via the coding length $L(\cdot)$ criteria. *Right*: CPP assigns semantic captions to each cluster via computing cosine similarities between text candidates and images and voting for the most suitable caption.

largest dataset containing more than 5 billion text-image pairs (Schuhmann et al., 2022). As a result, in this work, we adopt CLIP as our pre-trained model to design truly scalable and effective clustering learning algorithm.

**Image Clustering via Pre-trained Model.** The advance in vision pre-trained models have led to major breakthroughs in image clustering. SCAN (Van Gansbeke et al., 2020) proposes a three-step image clustering pipeline, starting with a self-supervised SimCLR (Chen et al., 2020) model, training a preliminary cluster head on the pre-trained features, and fine-tuning the trained cluster head via confidence-based pseudo-labeling. Subsequent research such as RUC (Park et al., 2021), TSP (Zhou & Zhang, 2022) and SPICE (Niu et al., 2022) has further enriched the field by exploring robust training, alternative network architecture, and different finetune methods. A few works (Tong et al., 2022a; Kim & Ha, 2021) also explored this paradigm in generative models, showcasing some promising downstream applications.

These pioneering methods have undoubtedly made remarkable progress. However, they often involve a degree of intricate engineering and parameter tuning. While these complexities are inherent to their design and have enabled them to achieve their goals, they may present challenges when implementing and scaling these methods to larger datasets. Recently, approaches like NMCE (Li et al., 2022) and MLC (Ding et al., 2023) have connected manifold clustering and deep learning via the MCR$^2$ principle (Yu et al., 2020; Ma et al., 2007a). In particular, MLC (Ding et al., 2023) has demonstrated that this principled approach has shown promise in terms of efficiency, scalability, and capability to handle imbalances present in larger datasets (Schuhmann et al., 2022; Lin et al., 2014) and real-life data. Consequently, we draw inspiration from MLC to develop a truly scalable and effective image clustering pipeline capable of handling the scale and natural imbalances present in real-world data.

## 3 OUR METHOD

We begin in §3.1 with a brief review of Manifold Linearizing and Clustering (MLC) (Ding et al., 2023), a framework that learns both a representation with desirable geometric properties and a clustering membership. In §3.2, we discuss how to effectively train MLC leveraging CLIP features as initialization. When the number of clusters is further not known, we describe how to identify the optimal number of clusters without costly retraining in §3.3. A few applications are left to §3.4: captioning the clusters and image-to-image search.

## 3.1 Review of Manifold Linearizing and Clustering

Given a dataset $\mathcal{X} = [\boldsymbol{x}_1, \ldots, \boldsymbol{x}_n] \in \mathbb{R}^{D \times n}$ of $n$ points lying on $k$ unknown manifolds, how to i) cluster the points in $\mathcal{X}$ and ii) learn a representation for $\mathcal{X}$ that has desirable geometric properties?

**Diverse and Discriminative Representation.** A fruitful line of research (Yu et al., 2020; Chan et al., 2020; Dai et al., 2022; Baek et al., 2022; Han et al., 2022; Li et al., 2022; Tong et al., 2022b), including MLC (Ding et al., 2023), considers learning a representation by using the principle of Maximal Coding Rate Reduction (MCR$^2$). Roughly speaking, an ideal representation pursued by MCR$^2$ should have the following properties.

• *Within-cluster diversity*: The features of each cluster spread well in a *low-dimensional linear subspace*. This naturally leads to a notion of (non-trivial) distance within a cluster, benefiting downstream tasks such as image retrieval (as shown in §4.2), generation, or interpolation.
• *Between-cluster discrimination*: Features (or subspaces) from different clusters are *orthogonal*. This is a typical goal in representation learning (Bengio et al., 2013), e.g., as pursued by the cross-entropy objective (Papyan et al., 2020; Zhu et al., 2021). Note that when each cluster is modeled by a low-dimensional linear subspace, this property further aids denoising in the feature space.

In particular, MLC seeks to find both a representation $\mathcal{Z} \in \mathbb{R}^{d \times n}$ of the data and a doubly stochastic clustering membership $\boldsymbol{\Pi} \in \mathbb{R}^{n \times n}$ where each entry $\Pi_{i,j} \geq 0$ measures the similarity between the $i$-th and $j$-th points. Here, $\mathcal{Z} = [f_{\boldsymbol{\eta}}(\boldsymbol{x}_1), \ldots, f_{\boldsymbol{\eta}}(\boldsymbol{x}_n)]$ where $f_{\boldsymbol{\eta}} : \mathbb{R}^D \to \mathbb{R}^d$ denotes a neural network with parameters $\boldsymbol{\eta}$, while $\boldsymbol{\Pi}$ is produced by a network parameterized by $\omega$ and a doubly stochastic projection (detailed below). MLC finds $\mathcal{Z}$ and $\boldsymbol{\Pi}$ by maximizing

$$\max_{\boldsymbol{\eta}, \boldsymbol{\omega}} \quad R(\mathcal{Z}(\boldsymbol{\eta}); \varepsilon) - R_c(\mathcal{Z}(\boldsymbol{\eta}), \boldsymbol{\Pi}(\boldsymbol{\omega}); \varepsilon), \tag{MLC}$$

$$\text{where} \quad R(\mathcal{Z}; \varepsilon) := \log \det \left( \mathcal{I} + \frac{d}{\varepsilon^2} \cdot \frac{1}{n} \mathcal{Z} \mathcal{Z}^\top \right), \tag{1}$$

$$R_c(\mathcal{Z}, \boldsymbol{\Pi}; \varepsilon) := \frac{1}{n} \sum_{j=1}^n \log \det \left( \mathcal{I} + \frac{d}{\varepsilon^2} \mathcal{Z} \operatorname{diag}(\boldsymbol{\Pi}_j) \mathcal{Z}^\top \right). \tag{2}$$

Broadly speaking, $R(\mathcal{Z}; \varepsilon)$ measures the volume of points in $\mathcal{Z}$ up to a $\varepsilon > 0$ rate distortion coding precision. Likewise $R_c(\mathcal{Z}, \boldsymbol{\Pi}; \varepsilon)$ measures the sum of volumes of the clusters encoded by $\boldsymbol{\Pi}$. Thus, MLC proposes to maximize the difference between the volumes, i.e., expand the volume of the embedded data globally, while compressing the volume of the embedded data within a cluster. This has been shown in (Yu et al., 2020) to provably learn a representation where features in each cluster (for a fixed $\boldsymbol{\Pi}$) spread well in a low-dimensional linear subspace, and subspaces from different clusters are orthogonal to each other.

**Doubly Stochastic Membership.** To learn a membership of the data for clustering, MLC draws inspiration from the success of doubly stochastic clustering (Lim et al., 2020; Ding et al., 2022) and computes a doubly stochastic membership from some latent codes of data. Specifically, MLC adopts a neural network $g_{\boldsymbol{\omega}} : \mathbb{R}^D \to \mathbb{R}^d$ with parameters $\boldsymbol{\omega}$ to first obtain latent codes $\mathcal{C} = [g_{\boldsymbol{\omega}}(\boldsymbol{x}_1), \ldots, g_{\boldsymbol{\omega}}(\boldsymbol{x}_n)]$ for each datapoint. Then, $\boldsymbol{\Pi}$ is given as a regularized projection $\operatorname{proj}_{\Omega, \gamma}(\mathcal{C}^\top \mathcal{C})$ onto $\Omega$, where $\Omega$ denotes the set of doubly stochastic matrices

$$\Omega := \{ \boldsymbol{\Pi} \in \mathbb{R}^{n \times n} : \boldsymbol{\Pi} \mathbf{1} = \boldsymbol{\Pi}^\top \mathbf{1} = \mathbf{1}; \quad \Pi_{ij} \geq 0, \quad \forall i, j \}. \tag{3}$$

Here, $\operatorname{proj}_{\Omega, \gamma}(\cdot)$ is defined as $\operatorname{argmin}_{\boldsymbol{\Pi} \in \Omega} -\langle \mathcal{C}^\top \mathcal{C}, \boldsymbol{\Pi} \rangle + \gamma \sum_{ij}^n \Pi_{ij} \log \Pi_{ij}$ which can be efficiently computed (Eisenberger et al., 2022; Sander et al., 2021), and $\gamma$ is a regularization strength that controls the entropy of $\boldsymbol{\Pi}$; in short, the larger $\gamma$ is, the more uniform $\boldsymbol{\Pi}$ is. Note that roughly speaking less uniform $\boldsymbol{\Pi}$ solutions result in fewer false connections between different clusters at the expense of less inter-cluster connectivity. We highlight here the dependency of $\mathcal{Z}$ and $\boldsymbol{\Pi}$ on their respective network parameters $\boldsymbol{\eta}, \boldsymbol{\omega}$, which we will typically omit for simplicity of notation.

## 3.2 Training MLC: Leveraging and Refining CLIP Features

**Structured Feature and Membership Initialization via CLIP.** Since (MLC) is non-convex, its initialization and optimization dynamics play a vital role in performance. Prior work (Li et al., 2022; Ding et al., 2023) used image self-supervised learning to initialize the features, which often fail to capture nuanced semantics, and as a result, they only reach, e.g., 33.5% clustering accuracy on

Tiny-ImageNet (Ding et al., 2023). We describe in the sequel how to initialize the representation and membership leveraging a pre-trained CLIP (Radford et al., 2021) model. Recall that CLIP has an *image encoder* and a *text encoder*, which maps input image and text respectively to a joint feature space. These encoders are trained utilizing billions of image-caption pairs. Motivated by the remarkable performance of CLIP in doing zero-shot tasks on unseen datasets, we take its pre-trained image encoder as our backbone (or feature extractor). The parameters of the backbone are henceforth fixed. Equivalently, we are taking the input data $\mathcal{X}$ to be CLIP features rather than raw images.

**Refining CLIP Features via MLC.** To allow fine-tuning of both $\mathcal{Z}$ and $\mathbf{\Pi}$, it is natural to add extra trainable layers after the backbone, as seen by the feature head and cluster head in Figure 1. However, these added layers could be arbitrary due to random initialization, undermining the quality of $\mathcal{Z}$ and $\mathbf{\Pi}$. Moreover, as seen in Figure 3, the pre-trained features from CLIP often have moderate pair-wise similarity even for those from very different classes. Toward this end, we propose to diversify the features $\mathcal{Z}$ simply via

$$\max_{\boldsymbol{\eta}} \quad R(\mathcal{Z}(\boldsymbol{\eta}); \varepsilon), \tag{4}$$

which is precisely (1); similar ideas have been explored also in (Li et al., 2022; Tong et al., 2023). In practice, we find that updating (4) only 1-2 epochs suffices to diversify $\mathcal{Z}$, making it a lightweight initialization step. To initialize $\mathbf{\Pi}$, we copy $\boldsymbol{\eta}$ to $\boldsymbol{\omega}$ (equivalently, assigning $\mathcal{C} = \mathcal{Z}$) without extra training effort, which is a benefit of using doubly stochastic membership. With both $\mathcal{Z}$ and $\mathbf{\Pi}$ initialized, we proceed and optimize all the added layers in (MLC). Once the training process is done, one can use the feature head and cluster head to obtain $\mathcal{Z}, \mathcal{C}$ for any set of images, seen or unseen; $\mathcal{C}$ in turn gives a $\mathbf{\Pi}$ following §3.1. When the number $k$ of clusters is determined or given, one can simply run spectral clustering (von Luxburg, 2007) on $\mathbf{\Pi}$ to get $k$ clusters.

## 3.3 Determining Number of Clusters without Retraining

In many scenarios, it is impossible for one to know the number of clusters. Two issues arise: i) one must have a mechanism to guess the number of clusters, and ii) to obtain an accurate estimate, one typically runs the entire deep clustering pipeline multiple times, which is computationally costly. In this regard, we propose to estimate the number of clusters without expensive retraining. This flexibility which alleviates ii) is attributed to the following fact: Note that the membership $\mathbf{\Pi} \in \mathbb{R}^{n \times n}$ merely signals pairwise similarity (recall §3.1), so the number $k$ of clusters is not part of the training pipeline of (MLC) whatsoever in contrast to the more common way of using a $n \times k$ $\mathbf{\Pi}$ matrix which encodes $k$ explicitly (Li et al., 2022). That said, given a $\mathbf{\Pi} \in \mathbb{R}^{n \times n}$, how can one know what is a reasonable number of clusters?

Towards this end, we again leverage the minimum coding length (or rate) criteria (Ma et al., 2007a), this time including not only the cost of features but also that of the *labels*. Recall the definition of *coding length* from (Ma et al., 2007a)

$$L(\mathcal{Z}) = (n + d)R(\mathcal{Z}; \varepsilon). \tag{5}$$

We now present Algorithm 1 to estimate the number of clusters. Assume that the training is done and one already has $\mathcal{Z}$ and $\mathbf{\Pi}$. For each $k \in \{1, \dots, K\}$ where $K$ is the max possible number of clusters, we do spectral clustering on $\mathbf{\Pi}$ to obtain $k$ clusters. Let $\mathcal{Z}_{[i]}$ denote the features from the $i$-th cluster and $|\mathcal{Z}_{[i]}|$ denote the number of features in $i$-th cluster. Then, (7) gives the cost $L_k$ of coding $k$ clusters, which includes the cost of the features of each cluster as well as the labels, which are the first and second terms in the summation. Finally, one can choose the optimal number of cluster that gives the lowest cost $L_k$.

## 3.4 Cluster Captioning and Image-to-Image Search

We end the section by noting that since MLC refines the CLIP features as well as performs clustering, several interesting applications could be done, such as captioning the learned clusters, as well as image-to-image search. For the interest of space, we showcase these applications in §4 and detail the procedures in Appendix H and Appendix G.

## 4 Experiments

In the sequel, we empirically verify the effectiveness of the CPP pipeline. We first compare CPP with state-of-the-art alternatives on CIFAR-10, -20, -100 (Krizhevsky et al., 2009) and ImageNet-1k

---

**Algorithm 1:** Clustering without knowing the number of clusters

---

**Input**: Learned features $\mathcal{Z} \in \mathbb{R}^{d \times n}$ and membership $\mathbf{\Pi} \in \mathbb{R}^{n \times n}$, max. # of clusters $K \in \mathbb{Z}_+$
For $k \leftarrow 1, \ldots, K$:

$$\mathcal{Z}_{[1]}, \ldots, \mathcal{Z}_{[k]} \leftarrow \text{Spectral clustering on } \mathbf{\Pi} \text{ to get } k \text{ clusters for } \mathcal{Z}; \qquad (6)$$

$$L_k \leftarrow \sum_{i=1}^{k} L(\mathcal{Z}_{[i]}) + |\mathcal{Z}_{[i]}| \left( -\log\left(\frac{|\mathcal{Z}_{[i]}|}{n}\right) \right). \qquad (7)$$

**Output**: Optimal number of clusters $\text{argmin}_k L_k$

---

Russakovsky et al. (2015). In §4.1, we show that CPP achieves higher clustering accuracy than deep clustering methods. In §4.2, we show that CPP learns a more structured representation than CLIP. We further challenge CPP to deal with large uncurated datasets, where the number of clusters is unknown, and the clusters are not annotated with captions. We report our findings on MS-COCO (Lin et al., 2014) and LAION-Aesthetics (Schuhmann et al., 2022) in §4.3 and WikiArt (Saleh & Elgammal, 2015) in §4.4. The full training details are left in Appendix C.

## 4.1 COMPARISON WITH DEEP CLUSTERING METHODS

**Methods.** We compare CPP with state-of-the-art deep clustering methods. These methods follow a similar two-stage approach as CPP, in that they leverage a pre-trained self-supervised or language-supervised network as initialization, and fine-tune the network with some clustering loss. In this subsection, all methods are given the ground-truth number of clusters. We refer the readers to Appendix A.2 for a full comparison with many more methods and their backbone architectures.

**Metrics.** For each method, given its estimated clusters, we compare them with the ground-truth ones to compute the clustering accuracy (ACC) and normalized mutual information (NMI).

| Method | CIFAR-10 | | CIFAR-20 | | CIFAR-100 | | ImageNet-1k | |
|---|---|---|---|---|---|---|---|---|
| | ACC | NMI | ACC | NMI | ACC | NMI | ACC | NMI |
| MLC (Ding et al., 2023) | 86.3 | 76.3 | 52.2 | 54.6 | 49.4 | 68.3 | - | - |
| SCAN (Van Gansbeke et al., 2020) | 88.3 | 79.7 | 50.7 | 48.6 | 34.3 | 55.7 | 39.9 | - |
| IMC-SWAV (Ntelemis et al., 2022) | 89.7 | 81.8 | 51.9 | 52.7 | 45.1 | 67.5 | - | - |
| TEMI* (Adaloglou et al., 2023) | 96.9 | 92.6 | 61.8 | 64.5 | 73.7 | 79.9 | 64.0 | - |
| CPP* | **97.4** | **93.6** | **64.2** | **72.5** | **74.0** | **81.8** | **66.2** | **86.8** |

Table 1: **Comparison with *state-of-the-art* deep clustering models**. Methods marked with an asterisk (*) uses pre-training from CLIP.

**Results.** From Table 1, it is evident that CPP outperforms alternatives in terms of ACC and NMI across all datasets; the same is true when CPP is compared with more methods in Appendix A.2. We observe that methods incorporating pre-training from external data have displayed significant improvement compared to their predecessors. This underlines the importance of integrating pre-trained models for image clustering. When compared with TEMI (Adaloglou et al., 2023), which also employs external pre-training data, CPP not only attains superior training accuracy but also drastically reduces training epochs: CPP converges within a maximum of 50 epochs, whereas previous methods typically require more than 100. This endows CPP with the capacity to efficiently and effectively scale up to even larger datasets in this era of pre-trained models. We will delve deeper into this in §4.3.

## 4.2 COMPARISON OF FEATURES LEARNED BY CLIP AND CPP

As CPP uses CLIP features as initialization, it is natural to ask: does CPP really improve the representation of CLIP? Below we answer this question in the affirmative: compared to CLIP features, CPP-learned features are i) qualitatively better structured, i.e., being within-cluster diverse and between-cluster discriminative, and ii)more amenable to tasks such as clustering and image-to-image search.

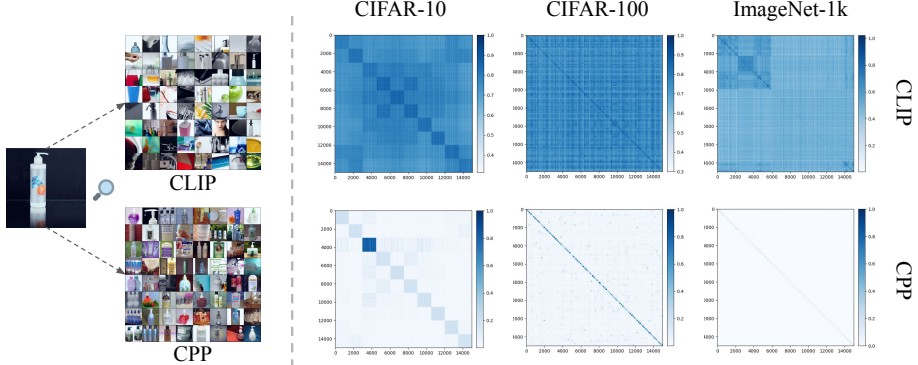

Figure 2: **Structured representations learned by CPP.** *Left*: An example of image-to-image search on ImageNet, using representations provided by CLIP (*Top*) and CPP (*Bottom*). *Right*: Cosine similarity $|\mathcal{Z}^\top \mathcal{Z}|$ visualization. Clear block-diagonal structures emerge in CPP-learned representations (*Bottom*), while the ones learned by CLIP show strong sample-wise correlation (*Top*).

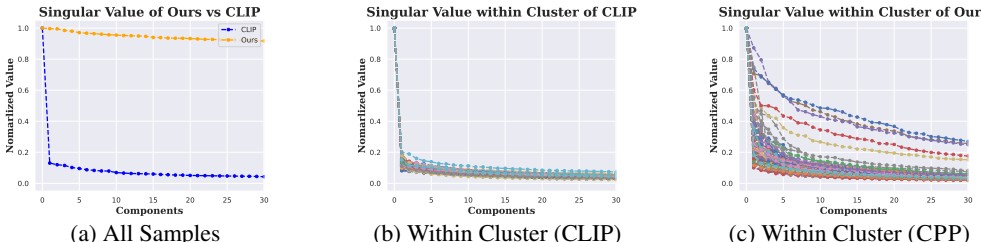

(a) All Samples      (b) Within Cluster (CLIP)      (c) Within Cluster (CPP)

Figure 3: **Normalized singular values of CIFAR-100 features.** *Left*: full dataset features. *Middle*: cluster-wise features from CLIP, membership given by KMeans. *Right*: cluster-wise features from CPP, membership given by spectral clustering upon membership matrix.

**Visualizing Cosine-Similarity of Features.** CPP aims at learning a union-of-orthogonal-subspace structure via optimizing the MLC (Ding et al., 2023) objective. To validate the effect of the training stage, we visualize the cosine similarity matrix of learned representations given by $|\mathcal{Z}^\top \mathcal{Z}|$ in the same manner of prior literature (Ma et al., 2007b). We observe that CPP's learned representations yield block-diagonal matrices in CIFAR-10/-100 and ImageNet-1k. In this setting, each block corresponds to one cluster, which qualitatively validates the effectiveness of CPP, i.e. representations within the same cluster are relatively similar, while those from different clusters are relatively dissimilar. In contrast, also in Figure 2, off-the-shelf CLIP representations do not exhibit similar structures.

**Visualizing Spectrum of Features by Cluster.** To better analyze the space structure of representations, we conduct PCA on the learned representations of CPP and CLIP. Specifically, we apply SVD on two groups of representations and visualize the normalized singular values. From Figure 3 (a), we observe that CLIP learns a highly-compacted space with features mainly spread around one direction (blue curve). In contrast, space learned by CPP shows to be more diverse with multiple dimensions (orange curve). Similar patterns appear in cluster-wise representations ( Figure 3 (b)(c)). Our results suggest that CPP learns holistically more diverse structures compared to the original CLIP.

**Measuring Clustering Performance.** Besides visualization, we evaluate the effectiveness of CPP through clustering tasks. As demonstrated in prior literature, CLIP (Radford et al., 2021) has already learned discriminative representations that can be employed for clustering. To assess the quality of these clusters without the application of CPP, and to validate the necessity of CPP, we apply KMeans, subspace clustering methods (EnSC (You et al., 2016a) and SSC-OMP (You et al., 2016b)), and spectral clustering on the representations learned through CLIP, comparing their results with CPP. From Table 2, we observe that CPP has achieved a significant improvement in cluster accuracy across all four datasets.

**Visualizing Image-to-Image Search Performance.** We further demonstrate the effect of structured representations on image-to-image search tasks. In this setting, we first establish an image repository

| Method | CIFAR-10 | | CIFAR-20 | | CIFAR-100 | | ImageNet-1k | |
|---|---|---|---|---|---|---|---|---|
| | ACC | NMI | ACC | NMI | ACC | NMI | ACC | NMI |
| KMeans | 83.5 | 84.0 | 47.3 | 51.3 | 52.3 | 67.7 | 49.2 | 81.3 |
| EnSC | 85.8 | 89.2 | 61.6 | 69.3 | 66.6 | 77.1 | 56.8 | 83.7 |
| SSC-OMP | 85.4 | 84.6 | 60.9 | 65.3 | 64.6 | 72.8 | 49.6 | 80.5 |
| Spectral Clustering | 73.6 | 75.2 | 52.4 | 56.2 | 67.2 | 76.1 | 55.8 | 83.4 |
| CPP | **97.4** | **93.6** | **64.2** | **72.5** | **74.0** | **81.8** | **66.2** | **86.8** |

Table 2: **Clustering accuracy and normalized mutual information** of classic clustering methods applied to CLIP features (*Top*) and the proposed CPP using CLIP as pre-trained features (*Bottom*). CLIP's ViT-L/14 is used as the backbone. See Appendix D for hyperparameter settings.

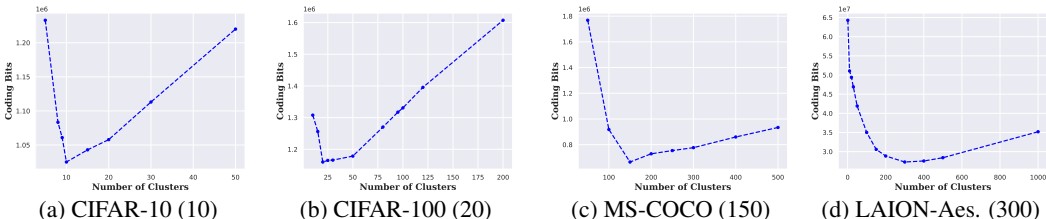

| (a) CIFAR-10 (10) | (b) CIFAR-100 (20) | (c) MS-COCO (150) | (d) LAION-Aes. (300) |

Figure 4: **Model selection for clustering** without knowing the number of clusters using Algorithm 1. For each dataset, the elbow point of the curve indicates the optimal number (in parenthesis) of clusters. The model selection is done efficiently *without* any retraining. See Appendix F for more results.

comprised of a set of images from the dataset, and then compute their respective representations, yielding our feature set. To search for a target image, we compute its representation and identify the 64 images from our feature set that have the closest cosine similarity to the target. We display one example in Figure 2, with additional results provided in Appendix G. Our results suggest that CPP representation entails a more comprehensive understanding, as demonstrated by the shampoo bottle example. Image-to-image search based on CPP recognizes the semantic meaning of a shampoo bottle as a daily hygiene product, while the search based on CLIP predominantly returns images of cylindrical objects.

### 4.3 CLUSTERING AND CAPTIONING ON LARGE UNCURATED IMAGE DATASETS

Recall that CPP is a complete clustering pipeline on large-scale datasets with specific designs for i) measuring the number of clusters (§3.3), and ii) captioning learned clusters (§3.4). In this subsection, we verify the designs on standard datasets CIFAR-10/-100 and study their effectiveness on datasets with larger scales such as MS-COCO and LAION-Aesthetics.

**Finding Optimal Number of Clusters.** We apply Algorithm 1 on the feature head representations after the training stage of CPP and report the results in Figure 4. Algorithm 1 identifies cluster numbers of 10 for CIFAR-10 and 20 for CIFAR-100, which aligns the finding in prior works (Van Gansbeke et al., 2020; Niu et al., 2022). Figure 4 also shows that the optimal number cluster for LAION-Aesthetics and MS-COCO are 300 and 150 respectively. We will use these two numbers to cluster LAION-Aesthetics and MS-COCO in the next paragraph.

**Captioning Clusters.** We apply Algorithm 2 in Appendix H on LAION-Aesthetics and MS-COCO. Notably, ideal text candidates for the stage would be a static high-quality vocabulary. In our setting, we utilize a mixture of LLM-generated labels and ImageNet-1k labels for efficiency, with details in Appendix H. Visualization of learned clusters and the corresponding captions can be found in Figure 5 and Appendix H. These results suggest that CPP's learned clusters can be summarized via consistent semantic meanings. CPP shows a promising qualitative performance on LAION-Aesthetics, even if it's large, diverse and imbalanced (see Figure 7 for details).

### 4.4 WIKIART: A CASE STUDY OF IMAGE CLUSTERING IN THE AGE OF PRE-TRAINING

In the age of pre-training, we now have easy access to general visual representations that span a broad spectrum of distributions. As a result, it is no longer necessary to undertake dataset-specific pre-training for tasks like image clustering. Building on this trend, CPP offers an efficient solution

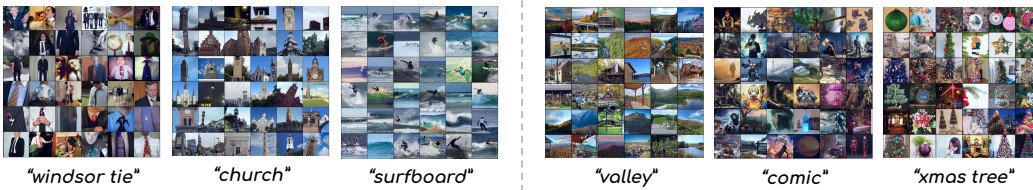

"windsor tie"    "church"    "surfboard"    "valley"    "comic"    "xmas tree"

Figure 5: **Examples of cluster captioning** on MS-COCO (*Left*) and LAION-Aesthetics (*Right*). More visualization can be found in Appendix H.

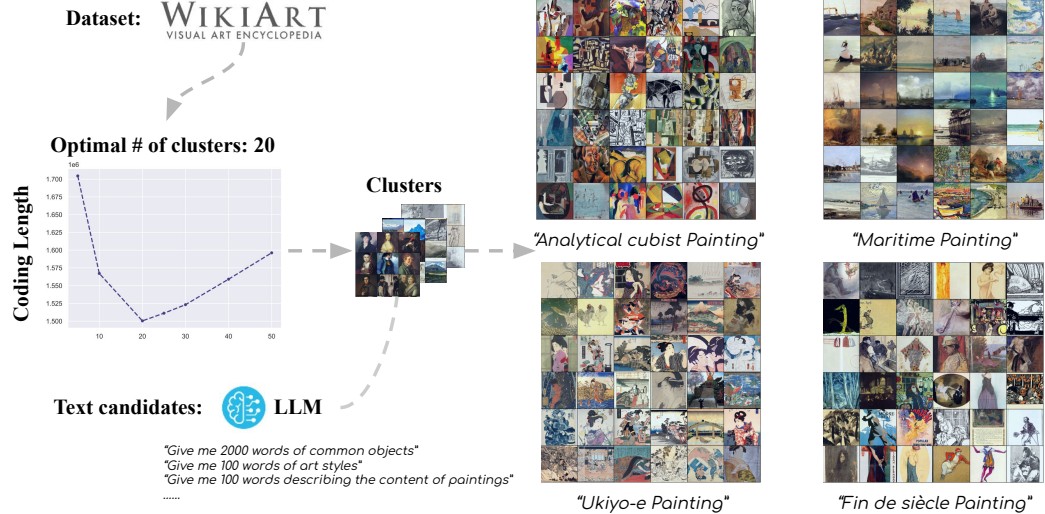

Figure 6: **Example of full CPP pipeline on WikiArt.** CPP clusters art pieces based on artistic and scenic styles and assigns them with meaningful captions from LLM-generated text candidates.

for clustering and captioning images from some rare distributions. We demonstrate this through a case study on WikiArt, a collection of art pieces from Wikipedia. Using CPP, we determined that 20 clusters optimally represent the dataset. We then captioned each cluster with a tailored description. Our findings reveal that CPP successfully categorized WikiArt into 20 distinct painting styles, including both scenic types like maritime and specific artistic movements such as analytical cubism. This capability makes CPP highly valuable for further analysis and application.

## 5 CONCLUSION AND DISCUSSION

This paper proposes a pipeline to do representation learning and clustering on large-scale datasets. The pipeline, dubbed CPP, takes advantage of CLIP pre-trained model. CPP achieves *state-of-the-art* clustering performance on CIFAR-10, -20, -100, and ImageNet-1k. Further, when the number of clusters is unknown, we give a mechanism for CPP that estimates the optimal number of clusters, without any costly retraining of deep networks. Finally, CPP refines the CLIP model, by giving more accurate clustering, as well as more diverse and discriminative representation, allowing better image-to-image search.

As for future work, we find it fascinating to explore the continual learning setting since real-world big data come only in a streaming fashion with new modes continuously showing up. It is also of interest to learn a diverse and discriminative representation and to cluster data with both text and image input.

## ACKNOWLEDGEMENTS

This work was partially supported by the Northrop Grumman Mission Systems Research in Applications for Learning Machines (REALM) initiative, DARPA Grant HR00112020010, NSF grant 1704458, and ONR MURI 503405-78051. Yi Ma acknowledges support from the joint Simons Foundation-NSF DMS grant #2031899, the ONR grant N00014-22-1-2102, TBSI, and support from the University of Hong Kong.

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

## A ADDITIONAL QUANTITATIVE RESULTS AND CLARIFICATIONS

### A.1 COMPARISON OF DIFFERENT PRE-TRAINED VISUAL MODELS

In our proposed pipeline (Figure 1), CLIP's image encoder is not the only candidate of pre-trained models. To further justify the generalizability of CPP, we evaluate the clustering performance of CPP leveraging visual representations from MAE (He et al., 2022) and DINO (Caron et al., 2021). For each model, CPP significantly improved the clustering performance and NMI score when compared with KMeans. Empirically, we find that both DINO and CLIP give good initializations for CPP; while MAE features are not suitable for image clustering, i.e. 12.3% accuracy via KMeans and 24.2% accuracy via CPP pipeline on ImageNet-1k, which is aligned with the discussion by Oquab et al. (2023) that MAE as a backbone are great for finetuning with labels, while less competent directly learn a discriminative representation.

| pre-train | Backbone | CPP Pipeline | | KMeans | |
|---|---|---|---|---|---|
| | | ACC | NMI | ACC | NMI |
| MAE | ViT L/16 | 24.2 | 69.8 | 12.3 | 60.6 |
| DINO | ViT B/16 | 59.0 | 84.5 | 53.5 | 81.6 |
| DINO | ViT B/8 | 61.9 | 86.8 | **56.0** | **85.2** |
| CLIP | ViT L/14 | **66.2** | **86.8** | 49.2 | 81.3 |

Table 3: **Benchmarking various models on ImageNet-1k with CPP and KMeans**. MAE and DINO models are pre-trained on ImageNet-1k; CLIP model is pre-trained on external data from their official implementation.

### A.2 COMPARISON WITH MORE DEEP CLUSTERING METHODS

In addition to Table 1, we list other *state-of-the-art* deep clustering methods and report the quantitative performance. The primary purpose of showing Table 1 and Table 4 is not to compete with or surpass other deep clustering methods. Instead, we aim to probe the boundaries of image clustering. We clearly list the backbones for reference.

| Method | Backbone | CIFAR-10 | | CIFAR-20 | | CIFAR-100 | | ImageNet-1k | |
|---|---|---|---|---|---|---|---|---|---|
| | | ACC | NMI | ACC | NMI | ACC | NMI | ACC | NMI |
| MLC (Ding et al., 2023) | ResNet-18 | 86.3 | 76.3 | 52.2 | 54.6 | 49.4 | 68.3 | - | - |
| SCAN (Van Gansbeke et al., 2020) | ResNet-18 | 88.3 | 79.7 | 50.7 | 48.6 | 34.3 | 55.7 | 39.9 | - |
| IDFD (Yaling Tao, 2021) | ResNet-18 | 81.5 | 71.1 | 42.5 | 42.6 | - | - | - | - |
| IMC-SWAV (Ntelemis et al., 2022) | ResNet-18 | 89.7 | 81.8 | 51.9 | 52.7 | 45.1 | 67.5 | - | - |
| RUC+SCAN (Park et al., 2021) | ResNet-18 | 90.3 | - | 54.3 | - | - | - | - | - |
| SPICE (Niu & Wang, 2021) | ResNet-34 | 91.7 | 85.8 | 58.4 | 58.3 | - | - | - | - |
| NMCE (Li et al., 2022) | ResNet-34 | 88.7 | 81.9 | 53.1 | 52.4 | - | - | - | - |
| TCL (Yunfan et al., 2022) | ResNet-34 | 88.7 | 81.9 | 53.1 | 52.9 | - | - | - | - |
| C3 (Sadeghi et al., 2022) | ResNet-34 | 83.8 | 74.8 | 45.1 | 43.4 | - | - | - | - |
| CC (Li et al., 2021) | ResNet-34 | 79.0 | 70.5 | 42.9 | 43.1 | - | - | - | - |
| ConCURL (Deshmukh et al., 2021) | ResNet-50 | 84.6 | 76.2 | 47.9 | 46.8 | - | - | - | - |
| Single-Noun Prior (Cohen & Hoshen, 2021) | ViT-B/32 | 93.4 | 85.9 | 48.4 | 51.5 | - | - | - | - |
| TEMI* (Adaloglou et al., 2023) | ViT L/14 | 96.9 | 92.6 | 61.8 | 64.5 | 73.7 | 79.9 | 64.0 | - |
| CPP* | ViT L/14 | **97.4** | **93.6** | **64.2** | **72.5** | **74.0** | **81.8** | **66.2** | **86.8** |

Table 4: **Comparison with more *state-of-the-art* deep clustering models**. Methods marked with (*) use pre-training from OpenAI CLIP, method use (#) use pre-training from OpenCLIP LAION-2B.

## B ABLATION STUDY

In this subsection, we conduct ablation studies on 2 components of CPP: pre-train datasets and diversified initialization.

The main results of our proposed methods leverage pre-training from OpenAI CLIP (Radford et al., 2021). To validate the generalizability of CPP, we additionally conduct experiments using ViT-L/14 from OpenCLIP (Ilharco et al., 2021), which is pre-trained on LAION-2B (Schuhmann et al., 2022). We report the results on Table 5.

| Backbone | Pretraining | CIFAR-10 | | CIFAR-20 | | CIFAR-100 | | ImageNet-1k | |
|---|---|---|---|---|---|---|---|---|---|
| | | CPP | KMeans | CPP | KMeans | CPP | KMeans | CPP | KMeans |
| ViT-L/14* | LAION-2B | 96.5 | 94.0 | 70.2 | 58.7 | 76.7 | 62.3 | 66.8 | 50.2 |
| ViT-L/14# | OpenAI | 97.4 | 83.5 | 64.2 | 47.3 | 74.0 | 52.3 | 66.2 | 49.2 |

Table 5: **Ablation study on pre-training**. We report the clustering accuracy and observe that CPP consistently outperforms KMeans on two different pretrained models. (*) leverages open source pretraining from OpenCLIP, (#) leverages pretraining from official CLIP.

We then conduct an ablation study on the contribution of diversified initialization (described in Section 3.2). We diversify the representation via optimizing the objective in equation (4). This procedure improves the clustering performance on various datasets with results reported in Table 6.

| Initialization | CIFAR-10 | | CIFAR-20 | | CIFAR-100 | | ImageNet-1k | |
|---|---|---|---|---|---|---|---|---|
| | ACC | NMI | ACC | NMI | ACC | NMI | ACC | NMI |
| Random | 87.6 | 90.6 | 57.4 | 69.9 | 67.9 | 78.3 | 63.7 | 84.7 |
| Diversified | **97.4** | **93.6** | **64.2** | **72.5** | **74.0** | **81.8** | **66.2** | **86.8** |

Table 6: **Ablation study on the contribution of diversified initialization.** We randomly initialize pre-feature layer, cluster head and feature head (*Top*) and compare the performance with the diversified initialization (*Bottom*) in equation (4).

## C TRAINING DETAILS

This section provides the training details - network architecture, datasets, optimization and hyperparameters.

**Datasets.** CIFAR contains $50,000$ training and $10,000$ test images, which are divided evenly into 10, 20 or 100 ground-truth classes, which we refer to as CIFAR-10, -20 or -100; note that the classes of CIFAR-20 are given by merging those of CIFAR-100. ImageNet incorporates around 1.2 million training images and $100,000$ test images, spread across $1,000$ classes, called ImageNet-1k. We process data in a manner identical to that used in CLIP (Radford et al., 2021), which involves resizing and center cropping images to dimensions of $224 \times 224$.

**Network Architecture.** We use a ViT L/14 model (Dosovitskiy et al., 2020) pre-trained via CLIP (Radford et al., 2021), with checkpoint from OpenAI[1]. As shown in Figure 1, we freeze the backbone during training and add a pre-feature layer composed of Linear-BatchNorm-ReLU-Linear-ReLU after the backbone. For feature head and cluster head, we use a Linear layer with mapping from the hidden dimension to feature dimension $d$ respectively. Unified architecture is applied on all the experiments across different datasets except adjusted hidden dimension and feature dimension $d$. Finally, to learn a ideal representation that spans a union of orthogonal subspaces as in §3.1, note that the feature dimension $d$ should be larger than or equal to the expected number of clusters. We leave more details in the Appendix C.

**Details of Added Layers.** For all datasets, we utilize a simple architecture composed of three parts: pre-feature layer, feature head and cluster head. Pre-feature layer has a structure with Linear-BatchNorm-ReLU-Linear-ReLU, with detailed setting in Table 7a. For feature head and cluster head, we use a linear layer respectively as is described in Table 7b.

**Dimensions.** Dimension $d$ and $d_{hidden}$ for each dataset are provided in Table 8a, note that $d$ should be larger than or equal to the expected number of clusters to satisfy the orthogonal subspace assumption.

**Optimizers.** We specify two independent optimizers to simultaneously optimize the MLC objective with detailed parameters in Table 8b.

---

[1] https://github.com/openai/CLIP

(a) Pre-feature layer

| |
|---|
| Linear: $\mathbb{R}^{768} \to \mathbb{R}^{d_{hidden}}$ |
| BatchNorm1d($d_{hidden}$) |
| ReLU |
| Linear: $\mathbb{R}^{d_{hidden}} \to \mathbb{R}^{d_{hidden}}$ |
| ReLU |

(b) Feature head and cluster head

| | |
|---|---|
| Feature head | Linear: $\mathbb{R}^{d_{hidden}} \to \mathbb{R}^{d}$ |
| Cluster head | Linear: $\mathbb{R}^{d_{hidden}} \to \mathbb{R}^{d}$ |

Table 7: Network Architecture

**Sinkhorn Distance.** The doubly stochastic membership matrix $\mathbf{\Pi}$ is computed by a sinkhorn distance projection on $\mathcal{C}^{\top}\mathcal{C}$, where the parameters $\gamma$ regulate the sparsity of the membership matrix as is described in Section 3.1. Details with this parameter are recorded in Table 8c.

**Optimization.** As describe in §3.2, we first warmup our network by 1-2 epochs by training $R(\mathcal{Z}; \varepsilon)$ alone, then simultaneously optimize both feature head and cluster head using (MLC). For both the feature head and cluster head, we train with SGD optimizer, learning rate set to 0.0001, momentum set to 0.9 and weight decay set to 0.0001 and 0.005 respectively.

**Initialization and Training Epochs.** Details in initialization (simply optimize $R(\mathcal{Z}; \varepsilon)$) epochs and total training epochs are recorded in Table 8d.

(a) Model Parameters. We adjust the dimension of learned features for the different expected numbers of clusters.

| Datasets/Parameters | $d$ | $d_{hidden}$ |
|---|---|---|
| CIFAR-10 | 128 | 4096 |
| CIFAR-20 | 128 | 4096 |
| CIFAR-100 | 128 | 4096 |
| ImageNet-1k | 1024 | 2048 |
| MS-COCO | 128 | 4096 |
| LAION-Aesthetics | 1024 | 2048 |

(b) Optimizers. We optimize the objective function using SGD optimizer with unified parameters as below:

| Optimizers | Type | lr | wd | momentum |
|---|---|---|---|---|
| Feature | SGD | 0.0001 | 0.0001 | 0.9 |
| Cluster | SGD | 0.0001 | 0.005 | 0.9 |

(c) Sinkhorn Distance Parameters while Training

| Datasets | $\gamma$ | Iter |
|---|---|---|
| CIFAR-10 | 0.175 | 5 |
| CIFAR-20 | 0.13 | 5 |
| CIFAR-100 | 0.1 | 5 |
| ImageNet-1k | 0.12 | 5 |
| COCO | 0.12 | 5 |
| LAION | 0.09 | 5 |

(d) Initialization epoch, total training epoch, batch size. Batch size doesn't affect too much on the performance. All experiments can be conducted on a single A100.

| Datasets | $epoch_{init}$ | $epoch_{total}$ | bs |
|---|---|---|---|
| CIFAR-10 | 1 | 5 | 1024 |
| CIFAR-20 | 1 | 15 | 1024 |
| CIFAR-100 | 1 | 50 | 1500 |
| ImageNet-1k | 2 | 20 | 1024 |
| COCO | 1 | 20 | 1200 |
| LAION | 2 | 20 | 1024 |

Table 8: Core hyperparameters selected in experiments.

## D    SUBSPACE CLUSTERING PARAMETERS

We conduct subspace clustering methods on CLIP features and report the highest accuracy after searching for optimal parameters.

**EnSC.** Both EnSC and SSC-OMP[2] estimate a membership matrix via solving some convex optimizations that depend only on CLIP features, and then run spectral clustering on the resulting membership. For EnSC, we use the efficient active-set solvers from (You et al., 2016a) to solve the convex optimization. EnSC has two parameters $\gamma, \tau$. Roughly speaking, $\tau \in [0, 1]$ balances between an $\ell_1$ and an $\ell_2$ penalty on the membership, with larger $\tau$ giving sparser affinity; $\gamma > 0$ is the weight of the data fidelity error, aside from the regularizing term.

**SSC-OMP.** $(k_{max}, \epsilon)$ We use the OMP solver for SSC (You et al., 2016b). $k_{max}$ is the maximum number of non-zero entries of each row of the membership, while $\epsilon$ controls the allowed data fidelity error.

**Spectral Clustering.** $\gamma$ denotes the parameter for sink horn distance projection, which is the same as the one mentioned in previous sections. For a given batch of CLIP's feature $\mathcal{C}' \in \mathbb{R}^{d \times n}$, we first normalize each feature vector and then do inner production plus sink horn distance projection, i.e. $\Pi_{CLIP} = \mathrm{proj}_{\Omega, \gamma}(\mathcal{C}'^\top \mathcal{C}')$. We then do spectral clustering on this membership matrix $\Pi_{CLIP}$.

Table 9: Parameter search with the following parameters for EnSC, SSC-OMP and spectral clustering. We report the highest performance on Table 2.

| Datasets | Parameters |
|---|---|
| EnSC | $\gamma \in [1, 5, 10, 50, 100], \tau \in [0.9, 0.95, 1.0]$ |
| SSC-OMP | $k_{max} \in [3, 5, 10], \epsilon \in [1e-4, 1e-5, 1e-6, 1e-7]$ |
| Spectral Clustering | $\gamma \in [0.2, 0.18, 0.16, 0.1, 0.09, 0.08, 0.07, 0.06]$ |

## E    EVALUATION ON IMBALANCED DATASETS

We evaluate CPP on imbalanced CIFAR-10 and imbalanced CIFAR-100, where images of odd classes (i.e. 1, 3, ...) are reduced to half of the original. Additionally, some large, uncurated datasets, like LAION-Aesthetic, exhibit natural imbalance. To demonstrate CPP's proficiency in identifying minority groups, we also present visualizations of clusters with fewer members in Figure 7.

## F    MORE RESULTS ON OPTIMAL NUMBER OF CLUSTERS

We additionally measure the coding length for ImageNet as is shown in Figure 8. For all datasets, we compute the coding length with $\epsilon^2 = 0.1$, which is consistent with the one in MLC objective function.

## G    IMAGE-TO-IMAGE SEARCH

**Pipeline.** Figure 9 demonstrates the pipeline of image-to-image search. In practice, the image repository is composed of 1.2M images from ImageNet's training split while the target image is randomly picked from ImageNet's validation split. We search the images in the repository via measuring the Euclidean distance and plot the 64 most similar images.

**Results.** Here, we provide 10 more image-to-image search results in Figure 10. We observe from these results that CPP learned better representation that facilitates image-to-image search.

---

[2]The implementations are provided by the authors at https://github.com/ChongYou/subspace-clustering.

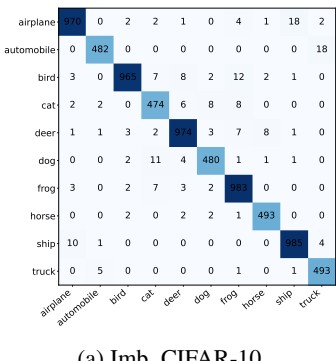
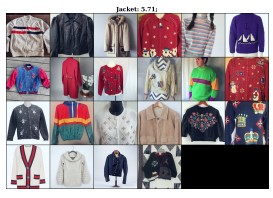
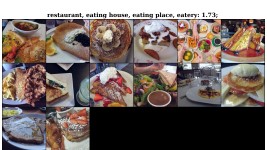

| (a) Imb. CIFAR-10 | (b) LAION-Aesthetic Cluster (i) | (c) LAION-Aesthetic Cluster (ii) |
|---|---|---|

Figure 7: Performance of CPP on imbalanced datasets. CPP achieved 97.3% and 71.3% clustering accuracy on Imb. CIFAR-10 and Imb. CIFAR-100 respectively; The confusion matrix demonstrates the prediction results on Imb. CIFAR-10 validation set (*Left*). LAION-Aesthetic is also a natural imbalanced dataset, where two clusters with few members are visualized, each composed of 0.73% and 0.47% images respectively from the dataset (clustering on 30k random samples from LAION-Aesthetic) (*Middle*, *Right*)

.

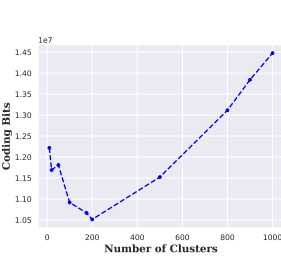

Figure 8: ImageNet (200)

# H   MORE RESULTS ON CLUSTERING AND LABELLING WITH TEXT

**Text Candidates Selection.** Ideally, an open vocabulary source with tons of highly reliable labels best suits our needs. However, text candidates of this quality and scale are usually hard to be obtained. In practice, In practice, we leverage powerful LLMs, such as GPT-4, to generate text candidates as an economical substitute. More specifically, we employ prompts like "generate 2000 names of real-world objects/creatures, generate 100 words of art styles, give me 100 words describning the content of paintings..." during the generation process. To guarantee the diversity and reliability, we furtherly mix them up with 1000 ImageNet class labels to construct the final 3000+ text candidates. It is noteworthy to mention that we did not utilize any label information from MS-COCO, LAION-Aesthetic or WikiArt.

**Pipeline.** We introduce a cluster-labeling algorithm after we obtain a well-trained CPP. First, we do spectral clustering upon the membership matrix given by CPP and get clusters of images. Then, for images in each cluster, we conduct weighted voting for the common labels. The voting algorithm is described in Algorithm 2.

**Results.** In this section, we visualize more cluster-captioning results for datasets including CIFAR-100, ImageNet-1k, COCO and LAION-Aesthetics. We also follow the optimal number of clusters measured in Section 4.3 for each dataset.

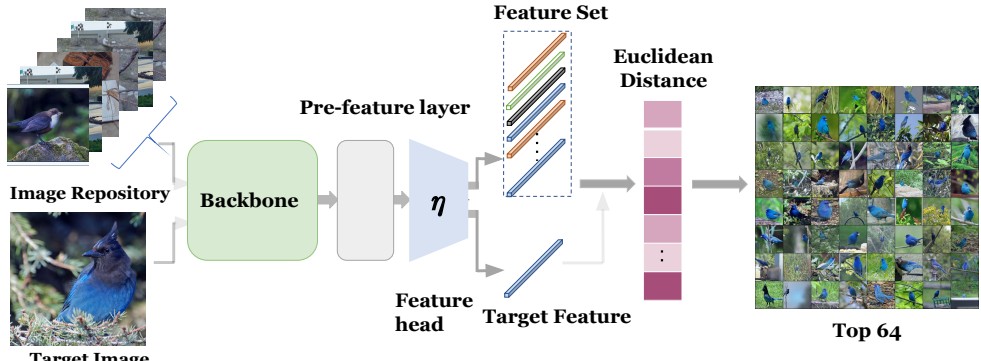

Figure 9: **Image-to-Image Search Pipeline.**

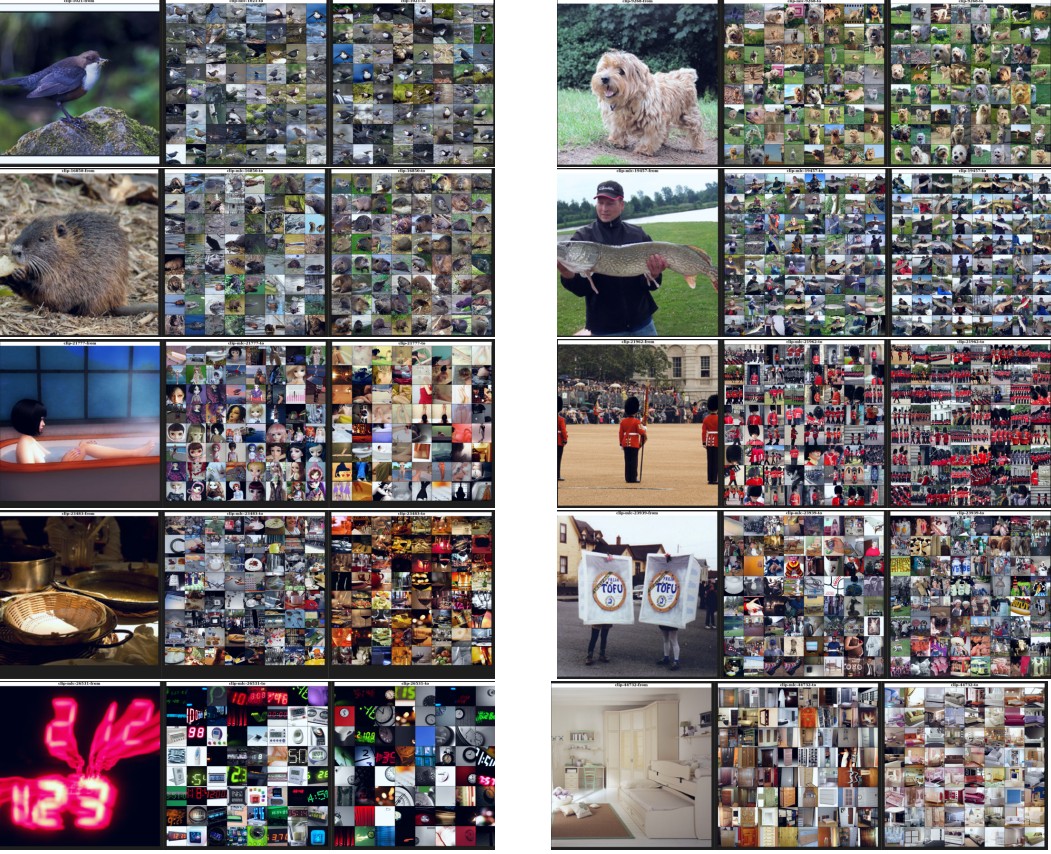

Figure 10: **Searching for similar images in ImageNet's training split.** *Left*: Target image from validation split in ImageNet. *Middle*: searched images via CPP's representation; *Right*: searched images via CLIP's representation.

---

**Algorithm 2:** Captioning one cluster

---

**Input**: Images from one learned cluster $\mathcal{X} \in \mathbb{R}^{N \times 3 \times 224 \times 224}$, $M$ text candidates, 0 initialized voting result vector $V \in \mathbb{R}^{M}$

$$\mathcal{Z}_{img} \leftarrow \text{CLIP: encode images}(\mathcal{X})$$
$$\mathcal{Z}_{txt} \leftarrow \text{CLIP: encode texts (text candidates)}$$

For $i \leftarrow 1, \ldots, N$:

$$\text{Scores4labels} \leftarrow \text{Cosine Similarity for}(\mathcal{Z}_{img}^{i}, \mathcal{Z}_{txt}) \quad (8)$$
$$\text{Valid Score} \leftarrow \text{Score4labels[top5]} \quad (9)$$
$$V \leftarrow V + \text{Valid Score} \quad (10)$$

**Output**: Caption for this cluster: text candidates$[\arg\max V]$

---

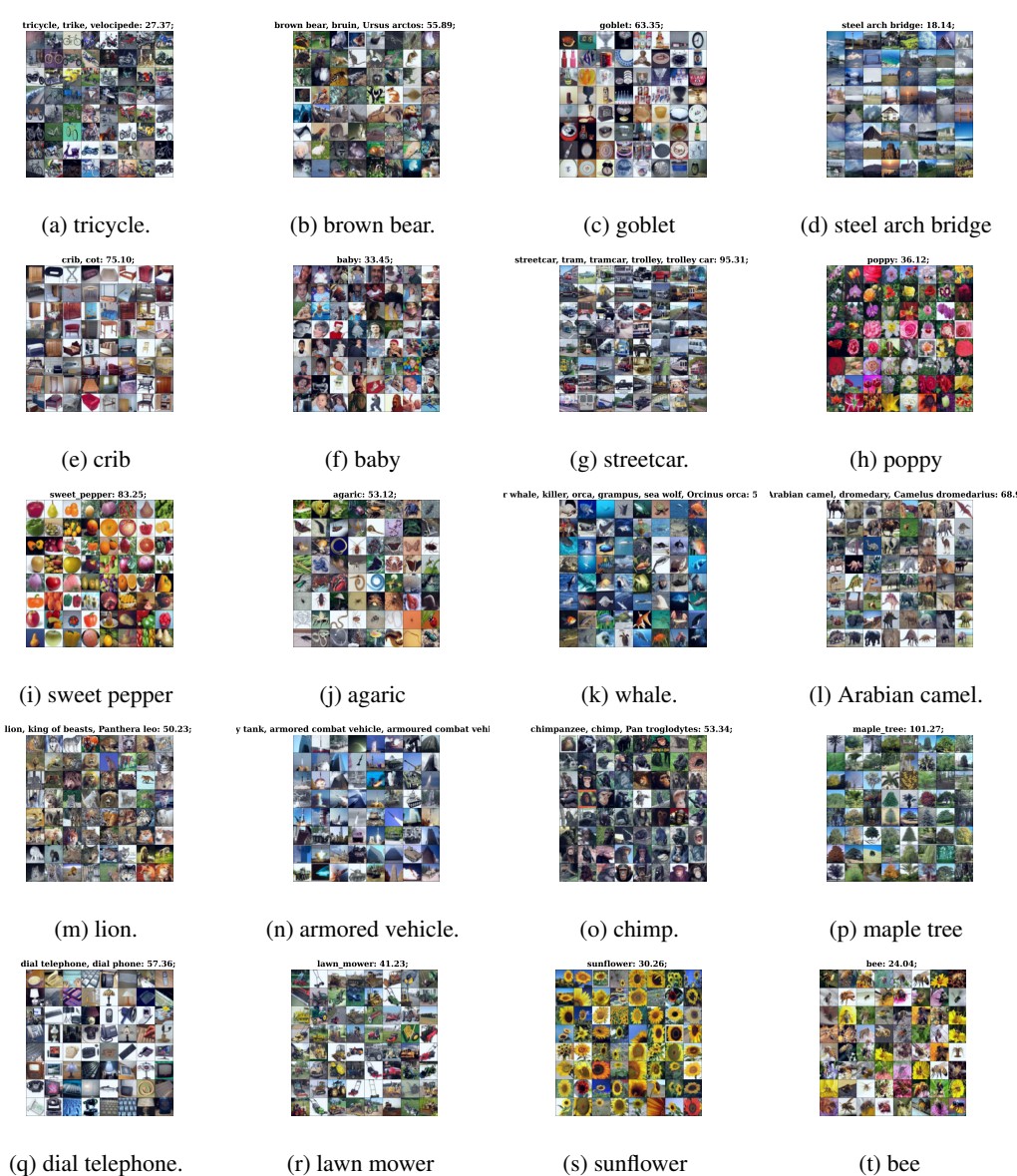

Figure 11: Clustering CIFAR-100 into 20 clusters and relabeling them using our pipeline.

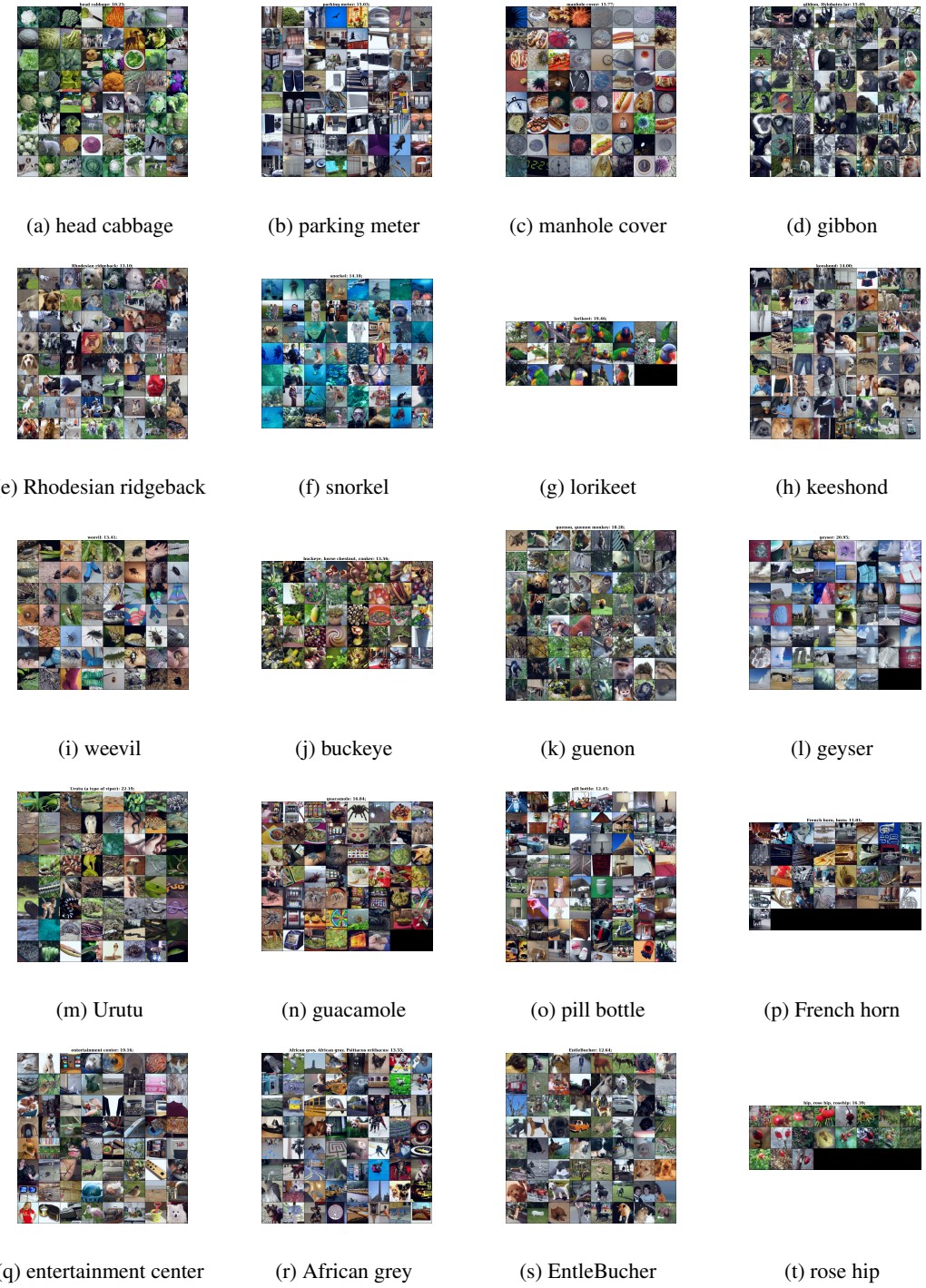

Figure 12: Clustering ImageNet (15k random samples from train split) into 200 clusters and relabeling them using our pipeline. (Randomly selected 20 clusters) Non-square figures represent that images within that cluster are not enough to fulfill the $8 \times 8$ grid.

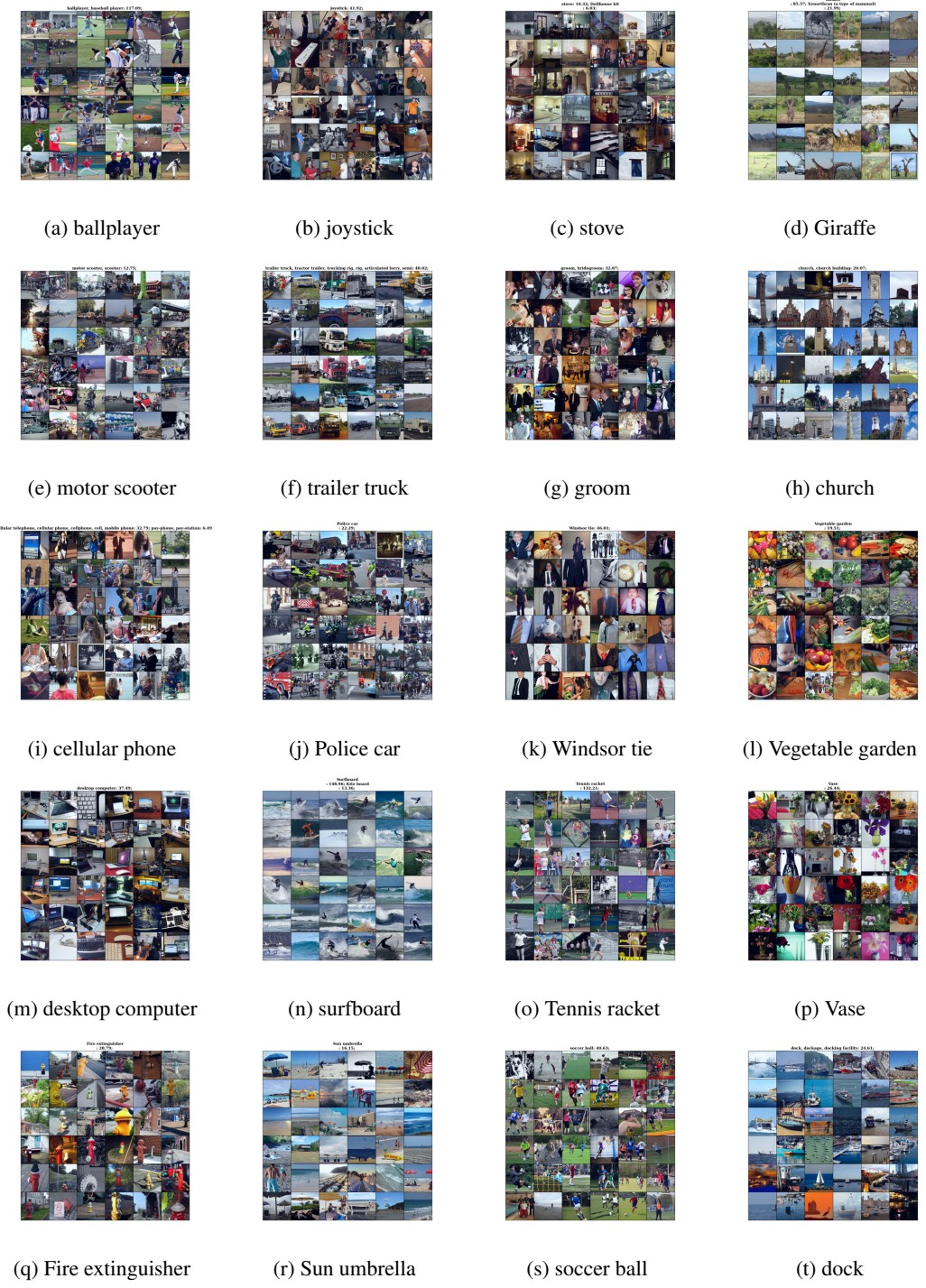

Figure 13: Clustering COCO (30k random samples) into 150 clusters and labeling them using our pipeline. (Randomly selected 20 clusters)

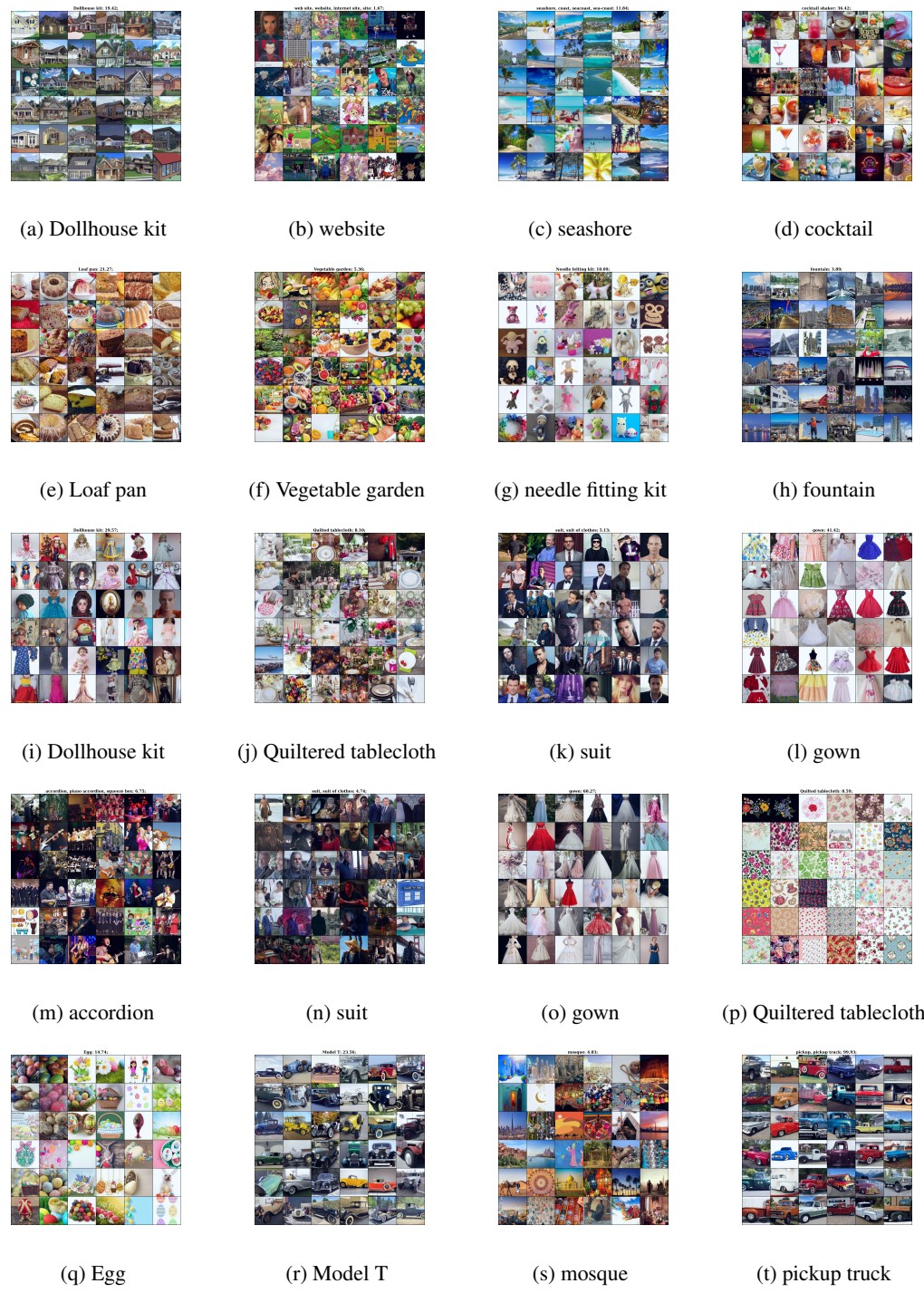

(a) Dollhouse kit     (b) website     (c) seashore     (d) cocktail

(e) Loaf pan     (f) Vegetable garden     (g) needle fitting kit     (h) fountain

(i) Dollhouse kit     (j) Quiltered tablecloth     (k) suit     (l) gown

(m) accordion     (n) suit     (o) gown     (p) Quiltered tablecloth

(q) Egg     (r) Model T     (s) mosque     (t) pickup truck

Figure 14: Clustering LAION-Aesthetic into 300 clusters and labeling them using our pipeline. (Randomly selected 20 clusters)

