# OpenReview forum: "Image Clustering via the Principle of Rate Reduction in the Age of Pretrained Models"
_ICLR.cc/2024/Conference — ICLR 2024 poster_

### Official Review · Reviewer_DGyy · 2023-10-29

**Soundness:** 3 good
**Presentation:** 3 good
**Contribution:** 3 good
**Rating:** 5
**Confidence:** 4

**Summary:**

This paper propose a new image clustering workflow that leverages the powerful feature representation capabilities of large pretrained models (like CLIP) to effectively and efficiently perform image clustering. It first develop a new algorithm to estimate the number of clusters in a given dataset. Through extensive experiments, the paper demonstrate that the workflow performs well on standard datasets.

**Strengths:**

1. Novel image clustering pipeline: The paper proposes a novel image clustering pipeline that leverages the powerful feature representation of large pre-trained models such as CLIP and cluster images effectively and efficiently at scale. The paper also develops a new algorithm to estimate the number of clusters in a given dataset, and a simple yet effective self-labeling algorithm that generates meaningful text labels for clusters.
2. State-of-the-art performance: The paper demonstrates that the proposed pipeline achieves state-of-the-art clustering performance on standard datasets such as CIFAR-10, CIFAR-100, and ImageNet-1k. The paper also shows that the pipeline works well on datasets without predefined labels, such as WikiArt

**Weaknesses:**

1. Dependence on pre-trained models: The paper heavily relies on the pre-trained models such as CLIP to provide the initial feature representation and the text candidates for labeling. The paper does not explore how the choice of pre-trained models affects the clustering performance or the quality of labels. The paper also does not consider the potential biases or limitations of the pre-trained models, such as their data sources, or domains.
2. Limited evaluation and comparison: The paper also does not report enough ablation studies or sensitivity analysis to show the impact of different components or hyperparameters of the pipeline.

**Questions:**

How does the choice of pre-trained models affect the clustering performance or the quality of labels?

---

> ### Author Response · Authors · 2023-11-16
> **Reply to Reviewer DGyy**
>
> Dear Reviewer`DGyy`:
>
> Thank you for the helpful comments and interesting questions! We appreciate that you find our work “novel” and attains “state-of-the-art”.  We respond to your questions below.
>
> >Dependence on pre-trained models: The paper heavily relies on the pre-trained models such as CLIP to provide the initial feature representation and the text candidates for labeling. The paper does not explore how the choice of pre-trained models affects the clustering performance or the quality of labels. The paper also does not consider the potential biases or limitations of the pre-trained models, such as their data sources, or domains.
>
> Thanks for pointing it out! We are with you that a more comprehensive discussion on the choice of pre-training solidifies our paper. Notably, we have conducted experiments using two additional popular pretrain models, MAE and DINO; please kindly see Table 3 in the Appendix, where we presented the observation in Appendix A.1. We also conducted experiments using OpenCLIP Table 9 in rebuttal supplementary material, which is pre-trained on open source data. Empirically, we find  that CLIP, OpenCLIP and DINO all lead to much higher clustering accuracy when used as pre-training for CPP.
>
> >Limited evaluation and comparison: The paper also does not report enough ablation studies or sensitivity analysis to show the impact of different components or hyperparameters of the pipeline.
>
> To address your concerns, We summarize the ablation studies we have conducted and added in this rebuttal period:
>
> 1. Our method improves as the pretrain model improves: we explore different pretraining: CLIP, MAE, DINO, and OpenCLIP. We have presented the results in Appendix A.1. and Table 9 in rebuttal supplementary material.
>
> 2. Diversified Initialization is important: we presented the ablation study of optimizing equation (4) in Appendix B.
>
> 3. Our method is  quite robust to hyperparameters: We additionally reported the effect of a different choice of $\gamma$ for sinkhorn distance projection. Please kindly refer to Appendix C.1. and Table 7(c) for definition, Table 10 in rebuttal supplementary material for additional results.
>
> **References**
>
> [1] Ding, Tianjiao, Shengbang Tong, Kwan Ho Ryan Chan, Xili Dai, Yi Ma, and Benjamin D. Haeffele. Unsupervised manifold linearizing and clustering. In Proceedings of the IEEE/CVF International Conference on Computer Vision (ICCV), pp. 5450–5461, October 2023.

---

> ### Author Response · Authors · 2023-11-21
> **Official Comment by Authors**
>
> Dear Reviewer DGyy,
>
> We are grateful for your efforts in the review, and we hope our response together with extra results covered your concerns. We are more than happy to further clarify or address additional questions. Please let us know if you still have any unclear parts of our work.
>
> Sincerely, Authors

---

> > ### Comment · Reviewer_DGyy · 2023-11-22
> >
> > Thank you for your response which has answered some of my concerns. I appreciate the considerable amount of experimental work that has been undertaken. However, I still have concerns about the fairness of the comparison with other clustering methods, since the big pretrained backbone (ViT-L) is used while most of other methods use much smaller backbones, like ResNet-18/34. Therefore, I will maintain my original score.

---

### Official Review · Reviewer_7qGL · 2023-11-01

**Soundness:** 4 excellent
**Presentation:** 3 good
**Contribution:** 3 good
**Rating:** 8
**Confidence:** 4

**Summary:**

The paper presents a unified approach for integrating feature learning with clustering processes. Its novel method for selecting the optimal number of clusters enhances the practicality of KNN and similar clustering methods for large-scale data, even when using computationally intensive models like CLIP. Additionally, it achieves state-of-the-art results on various datasets.

**Strengths:**

1- SOTA results
2- I find that enhancing KNN methods – whether by improving their scalability, representation, or explainability – is valuable.
3- Good visual analysis Fig 3 and 4.

**Weaknesses:**

1- The paper's flow is somewhat challenging to follow.
2- The acronym MLC is initially mentioned in the contributions section without prior definition, leading to initial confusion, though its relation to previous works becomes clearer later in the paper.
3- While the method appears to be a practical extension of MLC, its level of novelty and contribution to the field is not distinctly evident.

**Questions:**

1- The complexity added by Equation 4 to the optimization process isn't clear. Is it possible for there to be multiple optimal values for K?
2- Could a simpler method, like the elbow method, be used to determine K?
3- A brief explanation of what "more-structured representation" means in the context of related works would be helpful.

---

> ### Author Response · Authors · 2023-11-16
> **Reply to Reviewer 7qGL**
>
> Dear reviewer `7qGL`:
>
> Thank you for the helpful comments and interesting questions! We appreciate that you find  our work valuable and contains good analysis.  We respond to your questions below.
>
> >The acronym MLC is initially mentioned in the contributions section without prior definition, leading to initial confusion, though its relation to previous works becomes clearer later in the paper.
>
> Thanks for the suggestion! We will add a citation where the acronym shows up for the first time in the future revisions.
>
> >The complexity added by Equation 4 to the optimization process isn't clear. Is it possible for there to be multiple optimal values for K?
>
> **Complexity:** Overall, our pipeline has two parts: 1) learning the representation Z and membership Pi via optimizing equation (MLC), and 2) given learned Z and Pi, select the number of clusters, which is where Equation 4 appears. Part 1) is the computationally heavier one, as it involves training/finetuning a neural network with a large number of parameters (such as the transformers used in the paper). Part 2) is efficient, as it does not update the parameters of the neural network at all, and only involves spectral clustering on Pi (for which fast algorithms exist [5]) and evaluating Equation 6 (which takes O($d^2b$ ) to compute with b being the batch size).
>
> **Multiple Optimal Values for k:** You are right, in principle it is possible to have multiple optimal values of k.  For example, this could happen when the cost reduced by splitting a cluster into two is the same as that incurred by encoding one more label. However, empirically, as a result of complexity of real-world datasets, this did not happen in our experiments, as seen in Figure 4.
>
> >Could a simpler method, like the elbow method, be used to determine K?
>
> Good question! In the elbow method, one needs to compute the sum of squared distances from each point to the centroid of its assigned cluster. This is a good measure of cluster complexity only when the data from each cluster are close to a point. However, in our case, the data of each cluster are expected to lie close to a low-dimensional linear subspace, so the elbow method does not apply.
>
> >A brief explanation of what "more-structured representation" means in the context of related works would be helpful.
>
> Thanks for the suggestion! We originally wants to convey that “more-structured” means:
> - features within the same cluster tend to span a low-dimensional subspace (i.e., within-cluster diverse)
> - subspaces from different clusters tend to be orthogonal (between-cluster discriminative).
>
> Conceptually, learning such a representation is the goal of the rate reduction family[2, 3, 4], which has been mentioned in the introduction section. Empirically, we validate the claim in Figure 2, where we observed better image-to-image search and block-diagonal structures.
>
> Nevertheless, we are willing to refine the representation and explain it more clearly in the related works section in future revisions.
>
> >While the method appears to be a practical extension of MLC, its level of novelty and contribution to the field is not distinctly evident.
>
> We clarified the novelty and significance of the paper in the General Response above. Hopefully that alleviates your concern.
>
> **References**
>
> [1] Ma, Yi, Harm Derksen, Wei Hong, and John Wright. "Segmentation of multivariate mixed data via lossy data coding and compression." IEEE transactions on pattern analysis and machine intelligence 29, no. 9 (2007): 1546-1562.
>
> [2] Yu, Yaodong, Kwan Ho Ryan Chan, Chong You, Chaobing Song, and Yi Ma. "Learning diverse and discriminative representations via the principle of maximal coding rate reduction." Advances in Neural Information Processing Systems 33 (2020): 9422-9434.
>
> [3] Baek, Christina, Ziyang Wu, Kwan Ho Ryan Chan, Tianjiao Ding, Yi Ma, and Benjamin D. Haeffele. "Efficient maximal coding rate reduction by variational forms." In Proceedings of the IEEE/CVF Conference on Computer Vision and Pattern Recognition, pp. 500-508. 2022.
>
> [4] Ding, Tianjiao, Shengbang Tong, Kwan Ho Ryan Chan, Xili Dai, Yi Ma, and Benjamin D. Haeffele. Unsupervised manifold linearizing and clustering. In Proceedings of the IEEE/CVF International Conference on Computer Vision (ICCV), pp. 5450–5461, October 2023.
>
> [5] Yan, Donghui, Ling Huang, and Michael I. Jordan. "Fast approximate spectral clustering." Proceedings of the 15th ACM SIGKDD international conference on Knowledge discovery and data mining. 2009.

---

> ### Author Response · Authors · 2023-11-21
> **Official Comment by Authors**
>
> Dear Reviewer 7qGL
>
> We are grateful for your efforts in the review, and we hope our response together with extra results covered your concerns. We are more than happy to further clarify or address additional questions. Please let us know if you still have any unclear parts of our work.
>
> Sincerely, Authors

---

### Official Review · Reviewer_5jgz · 2023-11-01

**Soundness:** 3 good
**Presentation:** 3 good
**Contribution:** 2 fair
**Rating:** 3
**Confidence:** 4

**Summary:**

The submission introduces a new image clustering pipeline named CPP, which leverages large pre-trained models like CLIP to efficiently and effectively cluster images, particularly on large-scale datasets. The authors propose to estimate the optimal number of clusters in a dataset and optimize the rate reduction objective using pre-trained features, resulting in a notable improvement in clustering accuracy (e.g., from 57% to 66% on ImageNet-1k). Furthermore, by utilizing CLIP's multimodal capabilities, a simple yet effective self-labeling algorithm is developed to generate meaningful text labels for the clusters. The pipeline demonstrates state-of-the-art performance across various standard datasets including CIFAR-10, CIFAR-100, and ImageNet-1k, and extends its applicability to datasets without predefined labels like LAION-Aesthetics and WikiArts.

**Strengths:**

1. **Leveraging Large Pre-trained Models**: The integration of the powerful image encoder from CLIP into the clustering framework MLC significantly enhances the pipeline’s ability to process and analyze images, leading to state-of-the-art clustering performance on various datasets.

2. **Improvement in Clustering Accuracy**: Through the optimization of the rate reduction objective using pre-trained features, the pipeline achieves a noticeable improvement in clustering accuracy, as demonstrated on ImageNet-1k.

3. **Self-Labeling Algorithm**: The pipeline includes a simple yet effective self-labeling algorithm that leverages CLIP’s vision-text capabilities, resulting in semantically meaningful clusters that are comprehensible to humans.

**Weaknesses:**

1. **Limited Innovation in Methodology**: The main innovation of the proposed method seems to be centered around utilizing features extracted by CLIP for initialization, but there appears to be a lack of novelty in the algorithmic aspect of the approach.
2. **Concerns about Stability and Sensitivity**: As depicted in Fig.4, the vicinity of the extreme points in model selection appears quite flat, raising concerns about the algorithm's stability and its sensitivity to perturbations, such as the choice of hyperparameters and network architecture.
3. **Potential Information Leakage**: Given that CLIP has been trained on vast amounts of data, there is a suspicion of cluster/label information leakage in almost all of the experimental data (CIFAR and ImageNet) presented, which could potentially bias the quantitative results.
4. **Lack of Adequate Metrics for Text Labeling**: While the automated text annotation aspect of the pipeline is interesting, it seems to be lacking appropriate metrics to adequately evaluate and validate its performance.

**Questions:**

Please refer to the weakness.

---

> ### Author Response · Authors · 2023-11-16
> **Reply to Reviewer 5jgz**
>
> Dear Reviewer 5jgz:
>
> Thank you for the helpful comments and interesting questions! We appreciate that you recognize the contribution of our work (leverages pretrain models, improves cluster accuracy and proposes self-label algorithm).  We respond to your questions below.
>
> >Lack of Adequate Metrics for Text Labeling: While the automated text annotation aspect of the pipeline is interesting, it seems to be lacking appropriate metrics to adequately evaluate and validate its performance.
>
> CPP is one of the first methods to estimate text labels for **learned clusters** on datasets without ground-truth clusters or labels, thus we could not find existing metrics to evaluate. Still, we agree you have raised an interesting point.  To give some insights, we compute a simple metric for text labeling. We take the embeddings of images and the text label of each learned cluster using image and text encoders of CLIP ViT L-14, and compute the mean cosine similarity between image-text pairs. The mean similarity is around 0.22 to 0.3 for WikiArt and CIFAR-10. In contrast, irrelevant labels yield scores lower than 0.15. Our observation of the score frequency distribution aligns with Gadre et al[1]. For more quantitative and qualitative results, please kindly refer to Figure 15 in rebuttal supplementary material. We will summarize the results in the revised versions.
>
> >Concerns about Stability and Sensitivity: As depicted in Fig.4, the vicinity of the extreme points in model selection appears quite flat, raising concerns about the algorithm's stability and its sensitivity to perturbations, such as the choice of hyperparameters and network architecture.
>
> We’d like to clarify that:
>
> - Algorithm 1 is a post-hoc procedure after training. Hence, it will not impact our method’s stability.
>
> - Nevertheless, we think it crucial to examine the algorithm's stability in general. We summarize the ablation studies we have conducted and added in this rebuttal period below:
>
> 1. Our method improves as the pretrain model improves: we explore different pretraining: CLIP, MAE, DINO, and OpenCLIP. We have presented the results in Appendix A.1. and Table 9 in rebuttal supplementary material.
>
> 2. Diversified Initialization is important: we presented the ablation study of optimizing equation (4) in Appendix B.
>
> 3. Our method is  quite robust to hyperparameters: We additionally reported the effect of a different choice of $\gamma$ for sinkhorn distance projection. Please kindly refer to Appendix C.1. and Table 7(c) for definition, Table 10 in rebuttal supplementary material for additional results.
>
> >Potential Information Leakage: Given that CLIP has been trained on vast amounts of data, there is a suspicion of cluster/label information leakage in almost all of the experimental data (CIFAR and ImageNet) presented, which could potentially bias the quantitative results.
>
> We are with you on this point. Indeed, in the age of large pre-trained models, it is unlikely, if not impossible, that an image dataset publicly available for download has not been seen by pre-trained models. The only way to alleviate potential information leakage is to collect a private dataset that is available only for testing but not training, which we believe would be an interesting future contribution to the community.
>
> Nevertheless, we hope the following experiments reinforce the robustness of our evaluation.
>
> We presented additional quantitative evaluations of the clustering step in CPP leveraging:
>
> 1. DINO[2] ViT(Appendix A.1) pre-trained on pure visual data.
>
> 2. OpenCLIP[3](rebuttal supplementary material Table 9) pre-trained on LAION[4].
>
> There is less risk of "Information Leakage" for these models, since they were pre-trained in transparent settings of training strategy and data source. Notably, leveraging both of these models, CPP achieves competitive results on ImageNet-1k (DINO 61.9%, OpenCLIP 66.8%), with explicit improvement over KMeans. We hope these results prove the robustness of our evaluation and alleviate your concern.
>
> >Limited Innovation in Methodology: The main innovation of the proposed method seems to be centered around utilizing features extracted by CLIP for initialization, but there appears to be a lack of novelty in the algorithmic aspect of the approach.
>
> We clarified the novelty and significance of the paper in the General Response above. Hopefully that alleviates your concern.

---

> > ### Author Response · Authors · 2023-11-16
> > **References**
> >
> > [1] Gadre, Samir Yitzhak, Gabriel Ilharco, Alex Fang, Jonathan Hayase, Georgios Smyrnis, Thao Nguyen, Ryan Marten et al. "DataComp: In search of the next generation of multimodal datasets." arXiv preprint arXiv:2304.14108 (2023).
> >
> > [2] Caron, Mathilde, Hugo Touvron, Ishan Misra, Hervé Jégou, Julien Mairal, Piotr Bojanowski, and Armand Joulin. "Emerging properties in self-supervised vision transformers." In Proceedings of the IEEE/CVF international conference on computer vision, pp. 9650-9660. 2021.
> >
> > [3] Ilharco, Gabriel, Mitchell Wortsman, Ross Wightman, Cade Gordon, Nicholas Carlini, Rohan Taori, Achal Dave, et al. 2021. “OpenCLIP.” Zenodo. July. doi:10.5281/zenodo.5143773., https://github.com/mlfoundations/open_clip
> >
> > [4] Schuhmann, Christoph, Romain Beaumont, Richard Vencu, Cade Gordon, Ross Wightman, Mehdi Cherti, Theo Coombes et al. "Laion-5b: An open large-scale dataset for training next generation image-text models." Advances in Neural Information Processing Systems 35 (2022): 25278-25294.

---

> ### Author Response · Authors · 2023-11-21
> **Official Comment by Authors**
>
> Dear Reviewer 5jgz,
>
> We are grateful for your efforts in the review, and we hope our response together with extra results covered your concerns. We are more than happy to further clarify or address additional questions. Please let us know if you still have any unclear parts of our work.
>
> Sincerely, Authors

---

### Official Review · Reviewer_hovR · 2023-11-02

**Soundness:** 4 excellent
**Presentation:** 4 excellent
**Contribution:** 4 excellent
**Rating:** 8
**Confidence:** 5

**Summary:**

This paper proposed a novel method that leverages the rate reduction principle to learn to do image clustering using pretrained models. A technique for automatically select the optimal number of clusters is also proposed based on the same principle. Finally, a self-labeling mechism is proposed to label the clusters with semantic labels.
Experiments show that the proposed method achieves a good performance, as well as give a good estimation of the optimal number of clusters.

**Strengths:**

1. This paper provides an alternative way of performing image clustering, which seems to be performing well and could be of interesting for the community.
2. The method enables automatic estimation of the optimal number of clusters in a dataset, from the result the method seems to perform pretty well.

**Weaknesses:**

1. The main experiments are done on somewhat small datasets like CIFAR, or coarse grained dataset like COCO, the paper would be stronger if it could include finer-grained dataset for clustering like iNaturalist.

**Questions:**

1. I would be interesting in how the method perform on fine-grained datasets.
2. It would be better if the paper could include results of using other variant of CLIP models, such OpenCLIP.

---

> ### Author Response · Authors · 2023-11-16
> **Reply to Reviewer hovR**
>
> Dear reviewer `hovR`:
>
> Thank you for the helpful comments and interesting questions! We appreciate that you recognize that our work enables automatic estimation of optimal number of clusters and could be interesting for the community. We respond to your questions below.
>
> >It would be better if the paper could include results of using other variant of CLIP models, such OpenCLIP.
>
> Thank you for your valuable suggestion! We have conducted new experiments and reported the results in Table 9, rebuttal supplementary material. From our quantitative evaluation, we observed that OpenCLIP-ViT-L/14 achieved even better clustering accuracy on CIFAR-20, CIFAR-100, and ImageNet-1k compared with the original CLIP from OpenAI. It demonstrates great potential of our work, because it  improves as the pretrain model continues to evolve.
>
> >The main experiments are done on somewhat small datasets like CIFAR, or coarse grained dataset like COCO, the paper would be stronger if it could include finer-grained dataset for clustering like iNaturalist.
>
> Thank you for your suggestion! To clarify, we’ve already conducted experiments on large datasets such as LION-Aesthetic, which contains over 2.7 million images. Please kindly refer to our results in Figure 4, 5, and 14, where we observed that CPP performed well qualitatively, i.e. conceptually meaningful clusters with labels. Nevertheless, we appreciate your advice. We believe that a quantitative validation on finer-grained datasets will further solidify our results. Due to limited time in the rebuttal period, we will conduct experiments on iNaturalist and upload the results in future revisions.

---

> > ### Comment · Reviewer_hovR · 2023-11-20
> >
> > I would like to thank the authors for the response, it clarifies my concern. I would keep my positive score for this paper, and I would forward to the open source of the code of the paper

---

> > > ### Author Response · Authors · 2023-11-21
> > > **Thanks for Reply**
> > >
> > > Dear Reviewer hovR,
> > >
> > > Thank you for your kind reply and for maintaining your positive score for our paper. We are grateful for your interest and support of our work!
> > >
> > > Sincerely, Authors

---

### Official Review · Reviewer_2xo4 · 2023-11-07

**Soundness:** 3 good
**Presentation:** 3 good
**Contribution:** 2 fair
**Rating:** 5
**Confidence:** 4

**Summary:**

This paper studies the image clustering problem in the age of large pre-trained models. Specifically, this paper develops a method to determine the cluster number in a given dataset. Then, this paper validates that the features from large pretrained models, such as CLIP, help achive better custering accuracy than the traditional feature pre-training. Moreover, this paper also develops a self-labeling method to produce text labels for the clusters. Experiments on many image datasets, including ImageNet-1k, demonstrate the effectiveness of the proposed method.

**Strengths:**

- This paper achieves state-of-the-art results on many image datasets, including ImageNet-1k.

**Weaknesses:**

Three main technique contributions are developed in this paper, including a method to determine the cluster number, a validation that the features from CLIP push the limits of image clustering, and a self-labeling method to annotate the text-labels for the clusters. I have three concerns about these three technique contributions:

- This paper seems did not discuss the existing methods to determine the cluster number and the difference among them. Do all the existing clustering methods not discuss how to determine the cluster number?
- The proposed clustering method is a simple combination between CLIP features and MLC optimization method. I realize it is meaningful to validate the superiority of CLIP features in image clustering, but the technique contribution itself is kind of subtle.
- A self-labeling method to annotate the text-labels for the clusters in *Algorithm 2* simply uses a cosine similarity metric to determine which texts are the closest ones given text candidates, which is a very simple solution. It does not meet my expectations that the proposed self-labeling method strongly relies on the pre-define text candidates. What if the text candidates are not given?

**Questions:**

See the weaknesses.

---

> ### Author Response · Authors · 2023-11-16
> **Reply to Reviewer 2xo4**
>
> Dear reviewer `2xo4`:
>
> Thank you for the helpful comments and interesting questions! We appreciate that you recognize that our work achieved state-of-the-art on many datasets including ImageNet-1k. We respond to your questions below.
>
> >This paper seems did not discuss the existing methods to determine the cluster number and the difference among them. Do all the existing clustering methods not discuss how to determine the cluster number?
>
> Good question! The short answer to this question is No, existing deep clustering methods do not discuss how to determine the cluster number. We would like to clarify that 1) determining the number of clusters is a classical topic, 2) yet it is missing in modern deep clustering works.
>
> We go into further details below
>
> **Determining the number of clusters is a classical topic**
>
> The general paradigm to select the number of clusters, one first runs the clustering algorithm to estimate k clusters, then computes some score that measures how well the data fit the cluster model and how complex the cluster model is. Computing such scores is a well-studied topic: it can be done based on various assumptions on the geometric/statistical structures of the cluster models. Examples include the elbow method (as 7qGL also pointed out), Akaike/Bayesian Information Criterion.  We thank the reviewer for raising this question and we will add the discussion here to our revised version of the text.
>
> **Yet it is often missing in modern deep clustering works**
>
> Note, however, that one needs to estimate many cluster models, each corresponding to one choice of k. For most deep clustering works, this means retraining a deep network (or fine-tuning a cluster head) for each k, which is costly especially when data reaches ImageNet scale. Therefore, these works typically assume knowledge of the true number of clusters or an upper bound of it. Our method is different, because it learns the representation and membership (with deep networks) that do not rely on the number of clusters. This makes CPP suitable for real-world scale data.
>
> >A self-labeling method to annotate the text-labels for the clusters in Algorithm 2 simply uses a cosine similarity metric to determine which texts are the closest ones given text candidates, which is a very simple solution. It does not meet my expectations that the proposed self-labeling method strongly relies on the pre-defined text candidates. What if the text candidates are not given?
>
> Very interesting question! To clarify, we design Algorithm 2 in a time-efficient manner, as cosine similarity requires minimum computational cost. This design choice is partly inspired by the success of dual encoder-style VLMs such as CLIP[5]. These models have demonstrated their effectiveness in Image-Text retrieval due to their ability to pre-compute and store feature vectors. Leveraging this property, we can preprocess a large-scale open vocabulary of text candidates, thus alleviating the bias of candidate selection.
>
> >The proposed clustering method is a simple combination between CLIP features and MLC optimization method. I realize it is meaningful to validate the superiority of CLIP features in image clustering, but the technique contribution itself is kind of subtle.
>
> We clarified the novelty and significance of the paper in the General Response above. Hopefully that alleviates your concern.

---

> > ### Author Response · Authors · 2023-11-16
> > **References**
> >
> > [1] Van Gansbeke, Wouter, Simon Vandenhende, Stamatios Georgoulis, Marc Proesmans, and Luc Van Gool. "Scan: Learning to classify images without labels." In European conference on computer vision, pp. 268-285. Cham: Springer International Publishing, 2020.
> >
> > [2] Li, Zengyi, Yubei Chen, Yann LeCun, and Friedrich T. Sommer. "Neural manifold clustering and embedding." arXiv preprint arXiv:2201.10000 (2022).
> >
> > [3] Adaloglou, Nikolas, Felix Michels, Hamza Kalisch, and Markus Kollmann. "Exploring the Limits of Deep Image Clustering using Pretrained Models." arXiv preprint arXiv:2303.17896 (2023).
> >
> > [4] Ding, Tianjiao, Shengbang Tong, Kwan Ho Ryan Chan, Xili Dai, Yi Ma, and Benjamin D. Haeffele. Unsupervised manifold linearizing and clustering. In Proceedings of the IEEE/CVF International Conference on Computer Vision (ICCV), pp. 5450–5461, October 2023.
> >
> > [5] Radford, Alec, Jong Wook Kim, Chris Hallacy, Aditya Ramesh, Gabriel Goh, Sandhini Agarwal, Girish Sastry et al. "Learning transferable visual models from natural language supervision." In International conference on machine learning, pp. 8748-8763. PMLR, 2021.

---

> ### Author Response · Authors · 2023-11-21
> **Official Comment by Authors**
>
> Dear Reviewer 2xo4,
>
> We are grateful for your efforts in the review, and we hope our response together with extra results covered your concerns. We are more than happy to further clarify or address additional questions. Please let us know if you still have any unclear parts of our work.
>
> Sincerely, Authors

---

> > ### Comment · Reviewer_2xo4 · 2023-11-22
> > **Thanks for your rebuttal**
> >
> > Thanks for your rebuttal. I acknowledge the state-of-the-art clustering performance on large-scale datasets, and very appreciate it. Despire the proposed method pushes the limits of clustering, I am still concerned about the technique novelty, as mentioned by the authors that CPP indeed combines CLIP and MLC, and the experimental performance is just one aspect to evaluate a paper. For example, someone can exploit a more advanced CLIP-like pretrained model to combine with another existing state-of-the-art clustering method to further push the limits of image clustering. I would like to raise the rating from reject to borderline reject, and let AC to make the final decision.

---

### Author Response · Authors · 2023-11-16
**Global Response**

Thanks to all reviewers for their time and constructive comments. We look forward to engaging with the reviewers during the author-reviewer discussion period. Here, we summarize the reviews and address common concerns.

>Novelty and Significance

We are grateful that all reviewers appreciate the performance of the method, where reviewers also find the whole pipeline: novel(`DGyy`), interesting to the community(`hovR`), valuable in improving scalability, representation, and explainability(`7qGL`).

On the other hand, reviewer` 2xo4`, `hovR` and `7qGL` raised concerns about the clustering step in our method, as it combines CLIP and MLC.

We beg to disagree. It is true that the clustering step of CPP combines MLC and CLIP, however, the core contribution of this paper is to demonstrate that the complete CPP pipeline indeed pushes the limit of deep clustering. This is done by:

1. showing SOTA clustering performance on large-scale datasets such as ImageNet-1k and LAION (all reviewers find it as strengths of this paper);

2. designing novel and effective mechanisms to cluster datasets that have no labels nor number of clusters whatsoever, via a) estimating number of clusters, b) estimating semantic labels of obtained clusters (reviewer `hovR`, `5jgz`, `DGyy` find it as strengths)

3. refining image representations into target structure (within-cluster diverse and between-cluster discriminative), thus enhancing better image-to-image search. (reviewer `7qGL` finds it as a strength in visual analysis)

We emphasize that a *complete* clustering pipeline on large-scale datasets is lacking: (1) is omitted by most clustering works due to limited performance, and (2) has not been done in prior clustering works including MLC, since number of clusters is typically assumed to be known. In this sense, we believe that the contributions of our paper is significant.

---

> ### Author Response · Authors · 2023-11-16
> **Rebuttal Supplementary Material**
>
> For your reference, we list the pdf materials of this submission as follows:
>
> - `PDF` button: Full paper and Appendix, including Figure 1-14, Table 1-8.
>
> - `Supplementary Material` button: Rebuttal supplementary material, including Figure 15, Table 9-10.

---

### Meta-Review · Area_Chair_Pogs · 2023-12-03

**Metareview:**

The paper proposes an image clustering method based on fine-tuned visual representations from the visual-language pre-trained model CLIP and maximal coding rate reduction. It further proposes a method to estimate the number of clusters, which has proved to be a difficult task. The experimental results demonstrate considerable improvements over the clustering methods discussed in the paper.

Two out of four reviewers, Reviewer #hovR and Reviewer 7qGL, provided highly positive reviews. Both reviewers believe that the proposed method to estimate the number of clusters is novel, and the proposed image clustering method achieves SOTA results. Both reviewers indicated that they were satisfied with the authors' rebuttal on their comments.

Reviewer #2xo4 and Reviewer #DGyy provided slightly negative reviews but both acknowledge SOTA results demonstrated by the proposed clustering method. Reviewer #2xo4 believes that the pipeline of CLIP and MLC cannot be considered as a novelty. On the other hand, Reviewer #DGyy found the pipeline novel, but raised questions regarding the sensitivity of the pipeline on the pre-trained models (the authors provided additional experiments on different pre-trained models) and ablation studies (the authors provided ablation studies). The authors provided responses to the reviewers' comments, which I found satisfactory in general. Both reviewers acknowledged the authors' efforts during the rebuttal process but opted not to significantly change their ratings.

Although Reviewer #5jgz acknowledges SOTA results, their review was not in favor of the paper. Reviewer #5jgz thinks that the pipeline is not novel; however, they found the self-labeling algorithm to associate text labels to clusters interesting and innovative. The reviewer acknowledged the authors' efforts during the rebuttal process but opted not to change their rating.

In general, all reviewers acknowledge SOTA results demonstrated by the proposed clustering method. The majority also think that the proposed method to estimate the number of clusters is novel, and the self-labeling algorithm to associate text labels to clusters is interesting. They also acknowledged the authors' efforts during the rebuttal process. The AC thinks that the proposed pipeline obtained strong results in image clustering. Both the methodology and results have a broad impact on the ML community.

**Justification For Why Not Higher Score:**

It's possible to elevate the paper to a higher score, but the primary reason it's not higher is the scope of the paper. The improvement in clustering performance is attributed to the fine-tuned visual representations derived from the visual-language pre-trained model CLIP.

**Justification For Why Not Lower Score:**

The findings of this paper are interesting to the broad community where image clustering is needed.

---

### Decision · Program_Chairs · 2024-01-16

Accept (poster)